# Covariate Shift Corrected Conditional Randomization Test

**Bowen Xu**[*]
Harvard University
bowenxu@g.harvard.edu

**Yiwen Huang**[*]
Department of Statistics
Peking University
2000010773@stu.pku.edu.cn

**Chuan Hong**
Department of Biostatistics and Bioinformatics
Duke University
chuan.hong@duke.edu

**Shuangning Li**
Booth School of Business
University of Chicago
shuangning.li@chicagobooth.edu

**Molei Liu**[†]
Department of Biostatistics
Columbia Mailman School of Public Health
ml4890@cumc.columbia.edu

## Abstract

Conditional independence tests are crucial across various disciplines in determining the independence of an outcome variable $Y$ from a treatment variable $X$, conditioning on a set of confounders $Z$. The Conditional Randomization Test (CRT) offers a powerful framework for such testing by assuming known distributions of $X \mid Z$; it controls the Type-I error exactly, allowing for the use of flexible, black-box test statistics. In practice, testing for conditional independence often involves using data from a source population to draw conclusions about a target population. This can be challenging due to covariate shift—differences in the distribution of $X$, $Z$, and surrogate variables, which can affect the conditional distribution of $Y \mid X, Z$—rendering traditional CRT approaches invalid. To address this issue, we propose a novel Covariate Shift Corrected Pearson Chi-squared Conditional Randomization (csPCR) test. This test adapts to covariate shifts by integrating importance weights and employing the control variates method to reduce variance in the test statistics and thus enhance power. Theoretically, we establish that the csPCR test controls the Type-I error asymptotically. Empirically, through simulation studies, we demonstrate that our method not only maintains control over Type-I errors but also exhibits superior power, confirming its efficacy and practical utility in real-world scenarios where covariate shifts are prevalent. Finally, we apply our methodology to a real-world dataset to assess the impact of a COVID-19 treatment on the 90-day mortality rate among patients.

## 1   Introduction

Conditional independence tests are important across diverse fields for determining whether an outcome variable $Y$ is independent of a treatment variable $X$, conditioning on a potentially high-

---

[*]These authors contributed equally to this work.

[†]Corresponding author. To whom correspondence should be addressed.

38th Conference on Neural Information Processing Systems (NeurIPS 2024).

dimensional vector of confounding variables $Z$. This type of testing is critical for understanding the complex relationships among variables. For instance, scientists may hope to understand whether a specific genetic feature influences disease outcomes, whether a particular treatment effectively extends life expectancy, or whether certain demographic factors impact college admissions.

Traditionally, these conditional testing problems are approached by modeling $Y$ against $X$ and $Z$ through some parametric or semiparametric model. However, this strategy has been criticized due to potential model misspecification and limited observations of $Y$. As an alternative strategy, the model-X framework and Conditional Randomization Test (CRT) propose testing for the general conditional independence hypothesis $H_0 : X \perp\!\!\!\perp Y \mid Z$, free of any specific effect parameters [2]. The CRT assumes the distribution of $X \mid Z$ to be known and can control the type-I error exactly, allowing for the choice of any flexible, black-box test statistic. This strategy is particularly useful when there is either strong and reliable scientific knowledge of the distribution of $X \mid Z$ or an auxiliary dataset of $(X, Z)$ of large sample size, known as the semi-supervised setting.

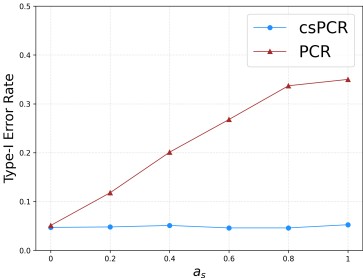

Figure 1: Type-I Error rates of our proposed csPCR and the source-only PCR on a simulated example. The Type-I error inflation of PCR demonstrates that source analysis is not valid or generalizable on the target due to covariate shift.

In practice, testing for conditional independence frequently involves using data from a source population to draw conclusions about a target population. This situation presents challenges due to potential differences in the distribution of variables between the two populations. For example, economists may be interested in whether college admission ($Y$) is independent of family income ($X$), conditioning on variables such as GPA, extracurricular activities, geographic location, and other demographics ($Z$). In the source population, the relationship might be influenced by factors like wealthy parents investing in SAT preparation, which boosts admission rates—a relationship that may not exist in a target population where such preparation is less common. Although $Y$ may not appear independent of $X$ given $Z$ in the source population, the conclusion could vary significantly in the target population. This discrepancy underscores the need for a robust and flexible testing procedure that can adapt to shifts in distributions.

More specifically, we address the *covariate shift* scenario, where the distributions of the treatment variables $X$, the confounding variables $Z$, and some surrogate or auxiliary variables $V$ (e.g., SAT scores) may differ between the source and target populations. However, the conditional distribution of $Y$ given $X$, $Z$, and $V$ remains the same between them. In such scenarios, our goal is to leverage information from the source to accurately test for conditional independence in the target population without the observation of $Y$ on target. In the scenario we consider, the presence of $V$ and potential differences in $P(V \mid X, Z)$ between the source and target populations may lead to the conditional independence $X \perp\!\!\!\perp Y \mid Z$ not holding simultaneously in the two populations. Specifically, because

$$P(Y \mid X, Z) = \int P(Y \mid X, Z, V) P(V \mid X, Z)\, dV,$$

the conditional distribution of $Y$ given $X$ and $Z$ can vary between populations. This underscores why the problem is non-trivial.

See Figure 1 for an example of the consequences of such covariate shift.

In this paper, we propose a novel conditional independence test suitable for covariate shift scenarios. Our method builds upon the Pearson Chi-Squared Conditional Randomization (PCR) test, a powerful model-X testing procedure that effectively addresses a broader range of alternative $p$-value distributions than the vanilla CRT [5]. Methodologically, we make two major contributions. First, we introduce importance weights into the label counting steps of the original PCR test, making the new test valid under covariate shift. These weights adjust the importance of each sample according to its density ratio, effectively rebalancing the source data to match the target population's distribution. Second, we introduce a power enhancement method that employs the control variates method to reduce variance in the test statistics. Although importance weights can increase the variance in test statistics, especially when the density ratio can become extremely high, potentially reducing power, our power enhancement method effectively addresses this issue. Together, these innovations enable us to develop a PCR test that is both powerful and valid under covariate shifts.

The rest of the paper is organized as follows: In Section 2, we provide a formal introduction to the problem setup. In Section 3, we introduce the proposed Covariate Shift Corrected Pearson Chi-squared Conditional Randomization (csPCR) test and establish that the proposed csPCR test controls the Type-I error asymptotically. In Section 4, we demonstrate the empirical performance of the csPCR test through simulation studies. In Section 5, we apply the proposed csPCR test to a real-world dataset to assess the impact of a COVID-19 treatment on the 90-day mortality rate among patients.

## 1.1 Related Work

Our work builds upon the model-X framework and the conditional randomization test proposed by Candes et al. [2]. The particular method we develop is based on a variant of the vanilla CRT, the Pearson Conditional Randomization (PCR) test [5]. Recent advances in the CRT include improving computation time [7, 10], studying robustness [6, 11], and examining statistical power [19]. The focus of this paper, different from the above, is on how to build a valid CRT procedure when there is covariate shift. The paper is also complementary to the above literature: for example, we hope that future work can conduct theoretical power analysis for our procedure or develop a double robust version of the procedure just like in [6]. Finally, we note that surrogate variables play a crucial role in this paper: because the distribution of the surrogate variables is different in the source and the target population, naively testing the conditional independence hypothesis in the source population can yield invalid conclusions for the target population. A surrogate or silver standard label is a variable that is more feasible and accessible than $Y$ in data collection and can be viewed as a noisy measure of $Y$. For example, tumor response rate is often used as an early endpoint surrogate for the long-term survival outcome [3], and blood pressure is commonly used as a surrogate for heart attacks. Surrogate variables are also commonly used in environmental studies and economics. Surrogate variables also play an important role in the paper by [6], albeit in a different way, where the surrogate variables are used to learn the distribution of $Y \mid X, Z$ and to further improve the robustness of the CRT procedure.

Statistical learning and inference under covariate shift has been extensively studied over the past years. As a seminal work in addressing covariate shift bias, [4] proposed a density ratio weighting approach using kernel mean matching to characterize the adjusting weights. Their key idea of importance (re)weighting is intrinsically connected with early work in broader contexts like importance sampling [15, e.g.] and semiparametric inference [13, e.g.]. [8] extended this idea to a doubly robust framework accommodating surrogate variables like $V$ and being more robust to the misspecification or poor quality of the density ratio models. [18] handled a more challenging scenario with severe shift and poor overlap between the source and target populations. Among this track of literature, [17] is the most closely related to our work as they also considered conditional independence testing under distributional shifts and proposed a general testing procedure base on importance sampling (IS) allowing for the use of CRT. Different from us, their work does not accommodate the covariate shifts of some surrogate or auxiliary $V$. Moreover, as will be shown in our numerical studies, their general IS testing strategy can encounter the loss of effective sample sizes and be less powerful than ours.

## 2 Problem Setup

### 2.1 Conditional Independence Testing under Covariate Shift

Let $Y \in \mathbb{R}$ denote the outcome variable, $X \in \mathbb{R}$ the treatment variable, $Z \in \mathbb{R}^p$ a vector of confounding variables, and $V \in \mathbb{R}^d$ a vector of surrogate variables. To make the problem more concrete, consider the following two examples:

**Example 1** (College Admission). *$Y$ is college admission, $X$ is family income, $Z$ includes a number of factors such as GPA, extracurricular activities, geographic location, and demographic information, $V$ is the SAT score. In this case, $V$ is easier to collect compared to $Y$ as the college admission requires individual-level surveys.*

**Example 2** (Health Outcome). *$Y$ is a long-term health outcome, $X$ is a medical treatment, $Z$ includes factors such as age, gender, and health history, $V$ includes surrogate variables like blood pressure, BMI, and duration of hospital stays post the treatment, which can be measured within a much shorter term than $Y$.*

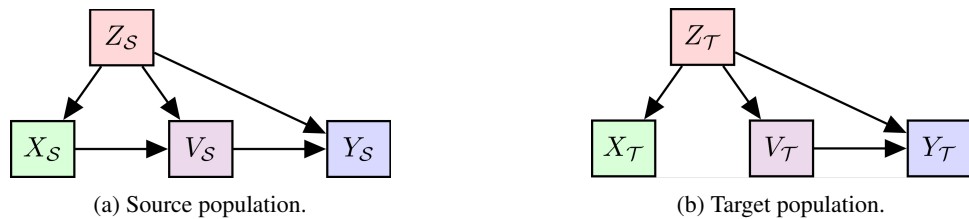

(a) Source population.                                    (b) Target population.

Figure 2: Direct acyclic graphs illustrating possible differences between the source and the target populations.

Consider a scenario involving two distinct populations: the source population $\mathcal{S}$ and the target population $\mathcal{T}$. We collect data from the source population with the goal of making inferences about the target population. The source data contains $n$ independent and identically distributed samples of $(Y_i, X_i, Z_{i\cdot}, V_{i\cdot})$ for $i = 1, \ldots, n$. Let $\mathbf{y} = (Y_1, Y_2, \ldots, Y_n)^\top \in \mathbb{R}^n$, $\mathbf{x} = (X_1, X_2, \ldots, X_n)^\top \in \mathbb{R}^n$, $\mathbf{Z} = (Z_{1\cdot}, Z_{2\cdot}, \ldots, Z_{n\cdot})^\top \in \mathbb{R}^{n \times p}$, and $\mathbf{V} = (V_{1\cdot}, V_{2\cdot}, \ldots, V_{n\cdot})^\top \in \mathbb{R}^{n \times d}$. We are interested in testing the following conditional independence hypothesis in the target population:

$$\mathcal{H}_0 : X \per\!\!\!\perp Y \mid Z. \tag{1}$$

We assume that the conditional distribution of $Y \mid X, Z, V$ is the same in both populations; however, the distribution of $(X, Z, V)$ varies between $\mathcal{S}$ and $\mathcal{T}$. More precisely, the joint distribution of $Y, X, Z, V$ can be described as follows:

$$\begin{aligned} P_{\mathcal{S}}(Y, X, Z, V) &= P_{\mathcal{S}}(X, Z, V) P(Y|X, Z, V) \quad \text{on } \mathcal{S}, \\ P_{\mathcal{T}}(Y, X, Z, V) &= P_{\mathcal{T}}(X, Z, V) P(Y|X, Z, V) \quad \text{on } \mathcal{T}. \end{aligned} \tag{2}$$

This situation is referred to as the *covariate shift* scenario because the distribution of the covariates $X, Z$, and $V$ in the source population $\mathcal{S}$ does not match that in the target population $\mathcal{T}$.

Let's understand the above assumption and its implications through the two examples above. In the college admissions example, it is plausible to assume that the rate of college admissions remains consistent across the two populations when conditioned on the SAT score, family income, and other confounding variables. However, the joint distribution of $X, V$ and $Z$ can differ: in the source population, if wealthy parents frequently invest in SAT preparation, boosting admission rates, this relationship may not hold in a target population where such preparation is uncommon. In such cases, it is thus possible that $X \not\!\perp\!\!\!\perp Y \mid Z$ in the source population but $X \perp\!\!\!\perp Y \mid Z$ in the target population (see Figure 2 for such an example). In the health outcomes example, it is again plausible that the conditional distribution of long-term health outcomes given the treatment variable, confounding variables, and surrogates remains the same across the two populations. However, the assignment of the treatment may depend differently on the surrogate variables across the two populations. Therefore, it's possible that $X \perp\!\!\!\perp Y \mid Z$ in one population, but not in the other.

In both examples, we can see that the result of naively applying a valid conditional independence test on the source population cannot guarantee a valid conclusion for testing $\mathcal{H}_0$ in the target population. Therefore, we need to develop new tools for addressing covariate shifts in conditional independence tests.

## 2.2 Model-X Framework

In this paper, we operate within the model-X framework, as described by Candes et al. [2], which assumes that the joint distributions of covariates $X, V, Z$ are perfectly known in both the source and target populations. This framework is particularly suited for scenarios where: (1) there is substantial prior domain knowledge about the covariates $X, V$, and $Z$, or (2) there is a significant amount of unsupervised data for these covariates in both populations, in addition to $n$ labeled observations in the source population, characterizing a semi-supervised setting.

An example of the first scenario can be seen in genetics, where researchers have well-established models for the joint distributions of single nucleotide polymorphisms (SNPs). For the second scenario, consider our earlier example involving health outcomes. Here, the outcome variable $Y$ represents a long-term health outcome that is more costly or sensitive to measure compared to the shorter-term

variables $X, V$, and $Z$. In such cases, the variables $X, V$, and $Z$ are typically easier and less costly to collect, frequently resulting in a semi-supervised setting in these health-related studies.

# 3  Method: Covariate Shift Corrected PCR Test

## 3.1  Incorporating the Density Ratio into the PCR Test

In Section 2.1, we discussed how naively applying conditional independence tests to the source data cannot guarantee valid conclusions for the target population. To address this issue, we must incorporate information about the differences between the two populations into our testing procedure. In particular, we will make use of the density ratio defined as:

$$e(X, Z, V) = \frac{P_{\mathcal{T}}(X, Z, V)}{P_{\mathcal{S}}(X, Z, V)}. \tag{3}$$

This ratio measures the relative likelihood of observing each combination of variables $(X, Z, V)$ in the target population compared to the source population. By reweighting the data points in the source population using this density ratio, we effectively transform the source distribution to match the distribution of the target population, thereby addressing the covariate shift problem.

More specifically, we build our method upon the recently proposed Pearson Chi-Squared Conditional Randomization (PCR) test [5]. Compared to the vanilla CRT, the PCR test is designed to be more powerful across a broader range of alternative $p$-value distributions. At a high level, the PCR test assigns a label to each data point following a counterfeit sampling step and a subsequent score computation step. Under the null hypothesis that $X \perp\!\!\!\perp Y \mid Z$, the distribution of these labels should be uniform across all possible labels. The PCR test then rejects the null hypothesis if the empirical distribution of the labels deviates significantly from uniformity, as determined by a Pearson's chi-squared test.

Under distributional shift, if the data points were sampled from the target population, then the distribution of the labels would be uniform. However, since the data points are actually sampled from the source population, they must be reweighted using the density ratio. More specifically, in the final step of the PCR test, where the Pearson's chi-squared test is applied, we consider not the count of data points for each label, but the sum of the density ratios of the data points for each label instead. Under the null hypothesis, each sum should approximate $n/L$, where $L$ is the total number of labels. Consequently, we modify the Pearson's chi-squared test to determine whether these weighted sums deviate significantly from $n/L$.

Based on the above intuition, we propose the Covariate Shift Corrected PCR (csPCR) Test, as outlined in Algorithm 1.

In Algorithm 1, lines 1-7 correspond to those in the original PCR test. These lines initiate the test by generating counterfeit samples $\tilde{X}_j^{(m)}$. Assuming the source and target populations were identical, under the null hypothesis, the random variables $(X_j, Y_j, Z_j), (\tilde{X}_j^{(1)}, Y_j, Z_j), \ldots, (\tilde{X}_j^{(M)}, Y_j, Z_j)$ would be exchangeable. Consequently, the rank $R_j$ would be uniformly distributed over $\{1, \ldots, M+1\}$ in the absence of ties, leading to a uniform distribution of the labels as well.

Lines 8-10 in Algorithm 1 address the covariate shift by incorporating density ratios as importance weights into $W_j$. Due to this redefinition of $W_\ell$, the null distribution of the final test statistic $U_{n,L}$ is also different. Therefore, we also adjust the rejection threshold from the quantile of a chi-squared distribution, as in the original PCR test, to the quantile of the weighted sum of chi-squared distributions.

## 3.2  Power Enhancement

To effectively address covariate shift, incorporating density ratios as importance weights into the PCR test is essential. However, when these ratios become large, they can increase the variance of the statistics $W_l$. This elevated variance can diminish the test's power. Therefore, developing methods to reduce this variance is crucial for maintaining the power of the test.

---

**Algorithm 1** Covariate Shift Corrected PCR (csPCR) Test.

---

**Input:** Data $D_{\mathcal{T}} = (\mathbf{y}, \mathbf{x}, \mathbf{Z}, \mathbf{V})$, the density ratio $e$, the test statistics $T$, integers $K, L \geqslant 1$, and the significance level $\alpha$.

1: Take $M = KL - 1$.
2: **for** each data point $j = 1$ to $n$ **do**
3:     Draw $M$ i.i.d samples $\widetilde{X}_j^{(1)}, \ldots, \widetilde{X}_j^{(M)}$ from $P_{\mathcal{T}}(X \mid \mathbf{Z})$.
4:     Use $T$ to score the initial data point $(X_j, Y_j, Z_j)$ and its $M$ counterfeits $(\widetilde{X}_j^{(1:M)}, Y_j, Z_j)$

$$
\begin{aligned}
T_j &= T(X_j, Y_j, Z_j) \\
\widetilde{T}_j^{(i)} &= T(\widetilde{X}_j^{(i)}, Y_j, Z_j), \text{ for } i \in \{1, \ldots, M\}.
\end{aligned}
\tag{4}
$$

5:     Let $R_j$ denote the rank of $T_j$ among $\{T_j, \widetilde{T}_j^{(1)}, \ldots, \widetilde{T}_j^{(M)}\}$, with ties broken randomly.
6:     Partition $\{1, \ldots, M+1\} = S_1 \bigcup \ldots \bigcup S_L$ with $S_\ell := \{(\ell-1)K + 1, \ldots, \ell K\}$. Assign label $\ell_j \in \{1, 2, \ldots, L\}$ to sample $j$ if $R_j \in S_{\ell_j}$.
7: **end for**
8: Let $w_j = e(X_j, Z_j, V_j)$ for each $j \in \{1, 2, \ldots, n\}$.
9: **for** each label $\ell \in \{1, 2, \ldots, L\}$: **do**
10:     Let $W_\ell$ be the sum of $\ell$-labeled importance weights: $W_\ell = \sum_{j=1}^{n} w_j \cdot \mathbb{1}\{\ell_j = \ell\}$.
11:     Let $D_\ell$ be the sum of $\ell$-labeled squared importance weights: $D_\ell = \sum_{j=1}^{n} w_j^2 \cdot \mathbb{1}\{\ell_j = \ell\}$.
12: **end for**
13: Let $\hat{\Omega}_n = \frac{L}{n}\mathrm{diag}(D_1, D_2, \cdots, D_L) - \frac{1}{L} \cdot \mathbf{1}_{L \times L}$.
14: Calculate the test statistic $U_{n,L}$ as follows $U_{n,L} = \frac{L}{n} \sum_{\ell=1}^{L} \left(W_\ell - \frac{n}{L}\right)^2$.

**Output:** Reject the null hypothesis if $U_{n,L} \geqslant \theta_{\hat{\Omega}_n, \alpha}$; otherwise, accept the null hypothesis. Here, $\theta_{\hat{\Omega}_n, \alpha}$ is the $1 - \alpha$ quantile of the distribution $\chi^2_{\hat{\Omega}_n}$, where $A \sim \chi^2_\Omega$ denotes that $A = x^\mathsf{T} x$ for $x \sim \mathcal{N}(0, \Omega)$.

---

To this end, we introduce a control variate function $a$, allowing $a(X, Z, V)$ to serve as a control variate in reducing variance in $W_l$ [14]. Specifically, for a chosen $\gamma_\ell$, we define

$$
\widetilde{W}_\ell = \sum_{j=1}^{n} w_j \cdot [\mathbb{1}\{\ell_j = \ell\} - \gamma_\ell a(X_j, Z_j, V_j)] + n\gamma_\ell \mathbb{E}_{\mathcal{T}} \left[a(X, Z, V)\right].
\tag{5}
$$

We can then use $\widetilde{W}_\ell$ instead of $W_\ell$ in our algorithm.

We note that for any arbitrary choice of the function $a$ and the parameter $\gamma_\ell$, the expectation of $\widetilde{W}_\ell$ would be the same as that of $W_\ell$:

$$
\begin{aligned}
\mathbb{E}\left[\widetilde{W}_\ell\right] &= \sum_{j=1}^{n} \mathbb{E}\left[w_j \mathbb{1}\{\ell_j = \ell\}\right] - \sum_{j=1}^{n} \gamma_\ell \mathbb{E}\left[w_j a(X_j, Z_j, V_j)\right] + n\gamma_\ell \mathbb{E}_{\mathcal{T}}\left[a(X, Z, V)\right] \\
&= \mathbb{E}\left[W_\ell\right] - n\gamma_\ell \left(\mathbb{E}_{\mathcal{S}}\left[e(X, Z, V)a(X, Z, V)\right] - \mathbb{E}_{\mathcal{T}}\left[a(X, Z, V)\right]\right) = \mathbb{E}\left[W_\ell\right].
\end{aligned}
\tag{6}
$$

Therefore, even if we make a sub-optimal choice of the function $a$ and the parameter $\gamma_\ell$ in practice, the resulting test (under certain assumptions) will still remain asymptotically valid (see Section 3.3 for more details).

However, for effective variance reduction, it is preferable to have the control covariates $a(X, Z, V)$ well-correlated with the outcome (See Section 4 for practical discussions on choices of the function $a$). This is quite feasible, especially since the surrogate variable $V$ is likely to be predictive of $Y$.

---

**Algorithm 2** Covariate Shift Corrected PCR Test with Power Enhancement.

---

**Input:** Data $D_{\mathcal{T}} = (\mathbf{y}, \mathbf{x}, \mathbf{Z}, \mathbf{V})$, the density ratio $e$, the test statistics $T$, the control variate function $a$, integers $K, L \geqslant 1$, and the significance level $\alpha$.

1: **for** each data point $j = 1$ to $n$ **do**
2:     Compute the labels $\ell_j$ as in Algorithm 1.
3: **end for**
4: Let $w_j = e(X_j, Z_j, V_j)$ for each $j \in \{1, 2, \ldots, n\}$.
5: **for** each label $\ell \in \{1, 2, \ldots, L\}$: **do**
6:     Compute $\hat{\gamma}_\ell$, the regression coefficient obtained by a weighted linear regression of the indicator function $\left\{\mathbb{1}\{\ell_j = \ell\}\right\}_{j=1}^n$ on the control variate $\left\{a(X_j, Z_j, V_j)\right\}_{j=1}^n$ with weights $\left\{w_j\right\}_{j=1}^n$.
7:     Compute the augmented version of $W_\ell$ as

$$\widetilde{W}_\ell = \sum_{j=1}^n w_j \cdot \left[\mathbb{1}\{\ell_j = \ell\} - \hat{\gamma}_\ell a(X_j, Z_j, V_j)\right] + n\hat{\gamma}_\ell \mathbb{E}_{\mathcal{T}}\left[a(X, Z, V)\right].$$

8: **end for**
9: Let $\mathbf{W} = \left(w_j \cdot \left[\mathbb{1}\{\ell_j = \ell\} - \hat{\gamma}_\ell a(X_j, Z_j, V_j)\right] + \hat{\gamma}_\ell \mathbb{E}_{\mathcal{T}}\left[a(X, Z, V)\right]\right)_{\ell,j}$ for $1 \leqslant \ell \leqslant L$, $1 \leqslant j \leqslant n$.
10: Calculate the sample covariance matrix $\widetilde{\Omega}_n = \frac{L}{n}(\mathbf{W} - \frac{1}{L} \cdot \mathbf{1}_{L\times n})(\mathbf{W} - \frac{1}{L} \cdot \mathbf{1}_{L\times n})^{\mathsf{T}}$.
11: Calculate the test statistic $U_{n,L}$ as follows $\widetilde{U}_{n,L} = \frac{L}{n} \sum_{\ell=1}^L \left(\widetilde{W}_\ell - \frac{n}{L}\right)^2$.

**Output:** Reject the null hypothesis if $\widetilde{U}_{n,L} \geqslant \theta_{\widetilde{\Omega}_n, \alpha}$; otherwise, accept the null hypothesis. Here, $\theta_{\widetilde{\Omega}_n, \alpha}$ is the $1 - \alpha$ quantile of the distribution $\chi^2_{\widetilde{\Omega}_n}$, where $A \sim \chi^2_\Omega$ denotes that $A = x^{\mathsf{T}}x$ for $x \sim \mathcal{N}(0, \Omega)$.

---

We would also like to discuss the choice of $\gamma_\ell$. According to the control covariate literature, with a fixed function $a$, the optimal choice of $\gamma_\ell$ that minimizes variance is given by:

$$\gamma_\ell = \frac{\mathrm{Cov}\left[w_j \mathbb{1}\{\ell_j = \ell\}, w_j a(X_j, Z_j, W_j)\right]}{\mathrm{Var}\left[w_j a(X_j, Z_j, W_j)\right]}. \tag{7}$$

This coefficient is also the same as that obtained from a linear regression [14]. Thus, when implementing the algorithm, we take $\gamma_\ell$ to be the regression coefficient obtained by running a weighted linear regression of the indicator function $\left\{\mathbb{1}\{\ell_j = \ell\}\right\}_{j=1}^n$ on the control variate $\left\{a(X_j, Z_j, V_j)\right\}_{j=1}^n$ with weights $\left\{w_j\right\}_{j=1}^n$.

We have outlined the new csPCR test, including this power enhancement step, in Algorithm 2.

### 3.3 Theoretical Properties

In this section, we establish that the proposed tests control the type-I error asymptotically. Furthermore, we show that the power enhancement step effectively reduces the variance of the statistics $W_\ell$, which can typically improve the power.

**Assumption 1** (Fourth moment). *The fourth moment of the density ratio $e(X, Z, V)$ is finite: $\mathbb{E}_{\mathcal{S}}\left[e(X, Z, V)^4\right] < \infty$. Furthermore, the fourth moment of product of the density ratio and the control variate function is also finite: $\mathbb{E}_{\mathcal{S}}\left[e(X, Z, V)^4 a(X, Z, V)^4\right] < \infty$.*

**Theorem 1** (Valid Tests). *Under Assumption 1, assume that the null hypothesis of $X \perp\!\!\!\perp Y \mid Z$ holds in the target population, then*

$$\lim_{n\to\infty} \mathbb{P}\left[\text{Algorithm 1 rejects}\right] = \alpha. \tag{8}$$

$$\lim_{n\to\infty} \mathbb{P}\left[\text{Algorithm 2 rejects}\right] = \alpha. \tag{9}$$

**Theorem 2** (Variance Reduction). *Let $W_l$ be the statistics computed in line 10 in Algorithm 1, and $\widetilde{W}_l$ be the statistics computed in line 7 in Algorithm 2. Under Assumption 1,*

$$\limsup_{n \to \infty} \left( \mathrm{Var}\left[\widetilde{W}_l\right] / \mathrm{Var}\left[W_l\right] \right) \leqslant 1. \tag{10}$$

## 4 Numerical Simulation

In this section, we present simulation studies to assess the performance of our proposed csPCR method and its power enhancement version denoted csPCR(pe), and compare them to a benchmark method. The benchmark method adopted is an importance-resampling based method [17], denoted as the IS method. For fair comparison, we used the same PCR statistic as our method for the testing with IS. We use a significance level of $\alpha = 0.05$.

### 4.1 Simulation Setup

We consider a semi-supervised setting where we have a large volume of unlabeled data of $(X_j, Z_j, V_j)$ from both the source and target populations. In addition, we have a small number of labeled data of $(Y_j, X_j, Z_j, V_j)$ from the source population.

We separate confounding variables $Z$ into two sets: $Z = (Z_{\mathrm{r}}, Z_{\mathrm{null}})$, where $Z_{\mathrm{r}}$ is the relevant set and $Z_{\mathrm{null}}$ is the null set. The relevant confounding variables $Z_{\mathrm{r}}$ are generated as i.i.d. multivariate normal, with mean 0 for the source population and 1 for the target population to simulate the distributional shift in $Z$, where $Z_{\mathrm{r}} \in \mathbb{R}^p$ and we set $p = 5$. Null confounding variables $Z_{\mathrm{null}}$ are generated independently with no correlation to other variables, modeled as $\mathcal{N}(0.1, I_q)$ with $q = 50$ for sparse high-dimensional settings in both populations.

The treatment variable $X$ and the surrogate variable $V$ are conditionally generated based on $Z$. Specifically, $X$ is modeled identically across both the source and target populations as $\mathcal{N}(u^\top Z_{\mathrm{r}}, 1)$, where $u$ is a predefined parameter vector that remains the same for both populations.

For $V$, it is modeled differently in the two populations, represented as $\mathcal{N}(v_{\mathcal{S}/\mathcal{T}}^\top Z_{\mathrm{r}} + (1-\theta)a_{\mathcal{S}/\mathcal{T}}X + \theta a_{\mathcal{S}/\mathcal{T}} \sin(X), 1)$. Here, $v_{\mathcal{S}}$ and $v_{\mathcal{T}}$ are predefined parameter vectors for the source and target populations, respectively. The parameter $a$ varies between populations ($a_{\mathcal{S}}$ for the source and $a_{\mathcal{T}}$ for the target), controlling the effect of $X$ on $V$, modeling the indirect effect. The factor $\theta$ modulates the nonlinear component of this relationship.

The outcome variable $Y$ is generated for both populations using the same conditional model over $(X, Z, V)$:

$$Y|(X, Z, V)_{\mathcal{S}/\mathcal{T}} \sim \mathcal{N}((v^\top Z_{\mathrm{r}})^2 + \beta V + \gamma X, 1),$$

where $\beta$ and $\gamma$ control the effects of $V$ (indirect) and $X$ (direct) on $Y$, respectively.

We generate 1000 unlabeled source and target samples to estimate the density ratio and generate 500 labeled source samples for testing. Moreover, in the simulation, we assume we have full knowledge of the joint distribution of $(X, Z)$ and estimate $V|X, Z$ using an Elastic net regression model with 5-fold cross-validation [20]. For the test statistic $T$ in the algorithm, we choose a simple function $T(\tilde{X}, Z, V, Y) = Y \cdot \tilde{X}$. For each parameter iteration, we conduct 1000 Monte Carlo simulations to estimate the Type-I error and power. We estimate the covariance matrix of the sequence of $W_i$'s using the Monte Carlo method and use the momentchi2 package [1] for calculating the $p$-value. Additionally, we empirically choose the best hyperparameter $L = 3$ for all our experiments through additional experiments shown in Appendix B.2.

### 4.2 Simulation Results

In Figure 3, we choose $a_{\mathcal{S}} = 1$ and $a_{\mathcal{T}} = 0$ to compare the Type-I error control of our methods with the benchmark. The left panel shows the Type-I error rate as the sample size of the data used to estimate the density ratio, $n_e$, varies from small to large. There appears to be a slight Type-I error inflation for all three methods when the sample size $n_e$ is small, but the Type-I error quickly converges to the ideal level of 0.05 as $n_e$ grows larger. Moreover, our methods show more stable Type-I error control than the benchmark method when the estimation sample size is low. The right

panel shows that when the density ratio is well approximated, all three methods attain good Type-I error control regardless of the change in $\beta$, i.e., the strength of the indirect effect, but the csPCR and csPCR(pe) methods have more stable control.

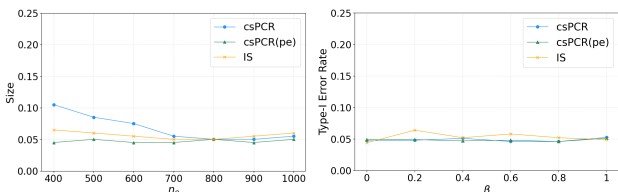

Figure 3: Comparison of Type-I error control across three methods.

To evaluate the statistical power of our csPCR test, we choose $a_\mathcal{S} = 0$ and $a_\mathcal{T} = 2$, so that the null hypothesis holds true in the target population but not in the source population. As Figure 4a shows, both the csPCR and the csPCR(pe) methods have uniformly higher power than the benchmark method as we vary the indirect

effect size $\beta$. For example, when $\beta = 1.4$, the benchmark IS method has a power of 0.33, the csPCR method has a power of 0.44, and the csPCR(pe) method can attain a power of 0.8.

When we fix the indirect effect $\beta = 2$ and vary the direct effect of $X$ ($\gamma$), as shown in Figure 4b, our methods still exceed the benchmark, and the power enhancement significantly improves the original version of the test. For example, when $\gamma = 1$, the benchmark IS method has a power of 0.4, the csPCR method has a power of 0.62, and the csPCR(pe) method can attain a power of 0.86.

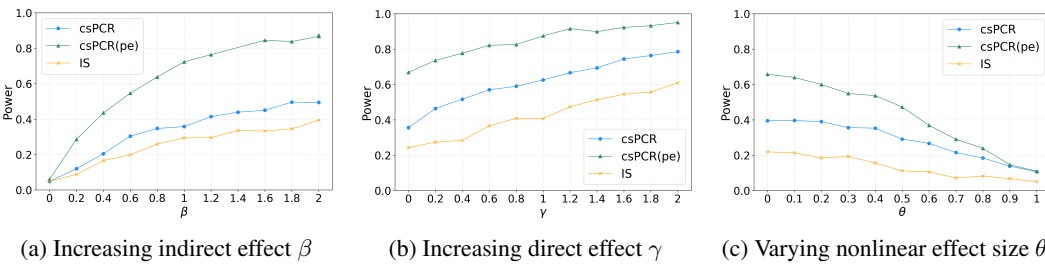

(a) Increasing indirect effect $\beta$  (b) Increasing direct effect $\gamma$  (c) Varying nonlinear effect size $\theta$

Figure 4: Comparison of statistical power of the three methods as the effect size varies: (a) indirect effect $\beta$, (b) direct effect $\gamma$, and (c) nonlinear effect size $\theta$.

We also test how adding a nonlinear component to the indirect effect affects the power when we assume a linear model of $V \mid Z, X$ in the estimation stage. This can be helpful in assessing the performance of our methods under model misspecification. As Figure 4c indicates, as the nonlinear effect increases, the power of all three methods decreases, though our methods still significantly exceed the benchmark. Interestingly, we observe that as $\theta \to 1$, i.e., there is a full nonlinear component without a linear component, the advantage of the power-enhanced version over the original csPCR test disappears. This occurs because when the $V \mid X, Z$ model is misspecified and the density ratio estimation is inaccurate, the variance reduction in the control variates step reduces variance in the "wrong" direction, and thus does not improve the power of the original method.

### 4.3 Effective Sample Size

We notice a series of work in measuring the effective sample size (ESS) of the density ratio reweighting approaches [9]. Among them, one of the most common measure is $n_{\text{eff}} = (\sum_{i=1}^{n} w_i)^2 / \sum_{i=1}^{n} w_i^2$. When the covariate shift between the source and target becomes stronger, the variance of the importance weight $w_i$ tends to be large and $n_{\text{eff}}$ will become smaller, which could result in lower power. We carry out simulation studies on the relationship between the power of csPCR and the ESS determined by the degree of covariate shift as discussed in Appendix B.3.

## 5 Real-World Application

The COVID-19 pandemic has presented unprecedented challenges to global health systems, with high variability in outcomes based on demographic and clinical characteristics. Early identification of patients at high risk for severe outcomes, such as mortality within 90 days of hospital admission, is crucial for timely and effective treatment interventions. This study leverages extensive hospital data to develop models predicting 90-day mortality following hospital admission due to COVID-19.

For this study, we extract patient data spanning from January 2020 to December 2023 from Duke University Health System (DUHS), focusing on individuals admitted with COVID-19. This period encompasses multiple waves of the pandemic, influenced by various circulating variants.

Our dataset comprises patient records for a total of $N = 3,057$ individuals admitted with COVID-19. The outcome $Y$ is defined as mortality within 90 days since hospital admission due to COVID-19. The treatment variable $X$ is defined as binary, where 1 indicates the administration of any COVID-19 specific medication (explained in Appendix C) and 0 otherwise. The covariates $Z$ include comorbidity indices (renal disease, diabetes without complication, diabetes with complication, local tumor, and metastatic tumor), age, gender, and race, which are critical for adjusting the risk models due to their known influence on COVID-19 outcomes. The length of hospitalization, denoted as $V$, is standardized to follow a standard normal distribution (with a mean of zero and a standard deviation of one), facilitating comparisons and integration into predictive models regardless of original scale or distribution.

The dataset is segmented into two distinct groups based on the date of hospital admission to align with pivotal changes in virus strain predominance and public health guidelines. The source data comprises COVID-19 admissions prior to November 30, 2021, with a sample size of $N_1 = 1,131$ patients. The target data includes admissions from November 30, 2021, through December 2023, totaling $N_2 = 792$ patients. This temporal division allows for the analysis of trends and outcomes associated with the evolving pandemic landscape. Prevalence of the 90-day mortality outcome within the source data is 14.3%, reflecting the impact of earlier virus strains and treatment protocols, while in the target data, the prevalence is substantially lower at 3.7%, possibly indicating the effect of improved treatments and vaccines, as well as the influence of different virus variants over time.

Table 1: $p$-values of different methods on COVID-19 dataset

| Method | csPCR | csPCR(pe) | IS |
|---|---|---|---|
| $p$-value | 0.025 | 0.032 | 0.663 |

For the analysis, we divide 50% of the source data, comprising 565 individuals, alongside the entirety of the target data, to estimate the density ratio. Density ratios of $X, Z$ are estimated using probabilistic classification method [12], while the density ratio of $V|X, Z$ is determined through Elastic Net regression. For all three methods, the test statistic $T$ is chosen to be $T(\tilde{X}, Z, V, Y) = Y \cdot \tilde{X}$. As indicated in Table 1, both csPCR and csPCR(pe) give statistically significant results, whereas the IS method does not. The statistically significant results are consistent with biomedical literature. For example, through systematic review and meta-analysis, [21] reported that Bamlanivimab is effective in reducing the mortality rates of COVID patients. In a cohort study, [16] also found similar effectiveness for Nirmatrelvir–ritonavir.

These results align with our findings from the simulation study and demonstrate that our method has increased power compared with the benchmark IS method.

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

# A Proofs

## A.1 Preliminaries

Throughout this section, we write $S(x_j, z_j, v_j) = s_j$ as the label assigned to sample $j$ in Algorithms 1 and 2, instead of using $\ell_j$. This notation helps avoid confusion between different label choices.

**Proposition 1.** *Assume that the conditional independence $X \perp\!\!\!\perp Y \mid Z$ holds on the target population $\mathcal{T}$. Let $e(x_j, z_j, v_j)$ denote the density ratio. For any integer $\ell \in [1, L]$, the following holds:*

$$\mathbb{E}_{\mathcal{S}}[e(x_j, z_j, v_j) \cdot \mathbb{1}\{S_{\mathcal{T}}(x_j, z_j, v_j) = \ell\}] = \frac{1}{L}.$$

*Proof of Proposition 1.* For simplicity, denote $w_j = e(X_j, Z_j, V_j)$ and $s_j = S_{\mathcal{T}}(X_j, Z_j, V_j)$.

$$
\begin{aligned}
\mathbb{E}_{\mathcal{S}}&[e(X_j, Z_j, V_j) \cdot \mathbb{1}\{S_{\mathcal{T}}(X_j, Z_j, V_j) = \ell\}] \\
&= \mathbb{E}_{\mathcal{S}}\left[\mathbb{E}\left[w_j \cdot \mathrm{P}(s_j = \ell \mid Y_j, Z_j, X_j, V_j)\right]\Big| Z_j, X_j, V_j\right] \\
&= \mathbb{E}_{\mathcal{S}}\left[w_j \cdot \mathbb{E}\left[\mathrm{P}(s_j = \ell \mid Y_j, Z_j, X_j, V_j)\right]\Big| Z_j, X_j, V_j\right] \\
&= \int_{(Z_j, X_j, V_j)} w_j \cdot p_{\mathcal{S}}(Z_j, X_j, V_j) \cdot \mathbb{E}\left[\mathrm{P}(s_j = \ell \mid Y_j, Z_j, X_j, V_j)\right] dZ_j \, dX_j \, dV_j \\
&= \int_{(Z_j, X_j, V_j)} p_{\mathcal{T}}(Z_j, X_j, V_j) \cdot \mathbb{E}\left[\mathrm{P}(s_j = \ell \mid Y_j, Z_j, X_j, V_j)\right] dZ_j \, dX_j \, dV_j \\
&= \mathbb{E}_{\mathcal{T}}\left[\mathbb{E}\left[\mathrm{P}(s_j = \ell \mid Y_j, Z_j, X_j, V_j)\right]\Big| Z_j, X_j, V_j\right] \\
&= \mathbb{E}_{\mathcal{T}}\left[\mathrm{P}(s_j = \ell \mid Y_j, Z_j, X_j, V_j)\right] \\
&= \frac{1}{L}.
\end{aligned}
\tag{11}
$$

The last equation follows from results in the non-covariate-shift scenario, e.g., from [5]. $\qquad\square$

## A.2 Proof of Theorem 1

### A.2.1 Results for Algorithm 1

Let $(W_\ell)_{\ell=1,\dots,L}$ be the sum of weights and $\hat{\Omega}_n$ be the sample covariance matrix in Algorithm 1. By Proposition 1, we have that

$$\mathbb{E}(W_\ell) = n \cdot \mathbb{E}[w_j \cdot \mathbb{1}\{\ell_j = \ell\}] = \frac{n}{L}.$$

Note that the $W_\ell$'s are sums of i.i.d. random variables, and thus by the Central Limit Theorem, as $n \to \infty$,

$$\mathbf{A}_n = \sqrt{\frac{L}{n}} \left(W_1 - \frac{n}{L}, W_2 - \frac{n}{L}, \dots, W_L - \frac{n}{L}\right) \xrightarrow{d} \mathcal{N}_L(0, \Omega),$$

where for any $\ell, \ell^* \in \{1, \dots, L\}$

$$
\begin{aligned}
\Omega_{\ell,\ell*} &= L\mathrm{Cov}(w_1 \mathbb{1}\{s_1 = \ell\}, w_1 \mathbb{1}\{s_1 = \ell^*\}) = L\mathbb{E}_{\mathcal{S}}\left[w_1^2 \cdot \mathbb{1}\{s_1 = \ell\}\mathbb{1}\{s_1 = \ell^*\}\right] - \frac{1}{L} \\
&= L\mathbb{E}_{\mathcal{S}}\left[w_1^2 \cdot \mathbb{1}\{s_1 = \ell\}\right] \mathbb{1}\{\ell = \ell^*\} - \frac{1}{L}.
\end{aligned}
$$

Therefore,

$$U_{n,L} = \mathbf{A}_n^{\mathsf{T}}\mathbf{A}_n \xrightarrow{d} \chi_\Omega^2.$$

Next, we will focus on the variance estimation part. We will show that $\hat{\Omega}_n \xrightarrow{p} \Omega$ as $n \to \infty$. For any $\ell, \ell^* \in \{1, \ldots, L\}$,

$$\hat{\Omega}_{n,\ell,\ell*} = \mathbb{1}\{\ell = \ell^*\} \frac{L}{n} D_l - \frac{1}{L} = \mathbb{1}\{\ell = \ell^*\} \frac{L}{n} \sum_{j=1}^{n} w_j^2 \cdot \mathbb{1}\{\ell_j = \ell\} - \frac{1}{L}$$

$$\xrightarrow{p} \mathbb{1}\{\ell = \ell^*\} L\mathbb{E}_{\mathcal{S}}\left[ w_1^2 \cdot \mathbb{1}\{s_1 = \ell\} \right] - \frac{1}{L} = \Omega_{\ell,\ell*}.$$

Up til now, we have that

$$U_{n,L} = \mathbf{A}_n^\mathsf{T} \mathbf{A}_n \xrightarrow{d} \chi_{\Omega}^2, \quad \text{and} \quad \theta_{\hat{\Omega}_n,\alpha} \xrightarrow{p} \theta_{\Omega,\alpha}.$$

Therefore,

$$\mathbb{P}(\text{Algorithm 1 rejects}) = \mathbb{P}(U_{n,L} \geqslant \theta_{\hat{\Omega}_n,\alpha}) \to \mathbb{P}(\chi_\Omega^2 \geqslant \theta_{\Omega,\alpha}) = \alpha.$$

### A.2.2 Results for Algorithm 2

Let $(\widetilde{W}_\ell)_{\ell=1,\ldots,L}$ and $\hat{\Omega}_n$ be the sum of weights and the sample covariance matrix in Algorithm 2. Let $\hat{\gamma}_\ell$ be the estimated coefficient in Algorithm 2.

Recall that in (7), we have identified the optimal choice of $\gamma_\ell$. We will start by working with this optimal choice and show that the $\hat{\gamma}_\ell$ is close to it. Define

$$\widetilde{W}_\ell = \sum_{j=1}^{n} w_j \left( \mathbb{1}\{\ell_j = \ell\} - \gamma_\ell a(X_j, Z_j, V_j) \right) + n\gamma_\ell \mathbb{E}_{\mathcal{T}}[a(X, Z, V)],$$

$$K_{\ell,j}(\gamma) = w_j \left( \mathbb{1}\{\ell_j = \ell\} - \gamma a(X_j, Z_j, V_j) \right) + \gamma\mathbb{E}_{\mathcal{T}}[a(X, Z, V)], \text{ and}$$

$$H_j = w_j a(X_j, Z_j, V_j) - \mathbb{E}_{\mathcal{T}}\left[ a(X, Z, V) \right].$$

Therefore, we have $\widetilde{W}_\ell = \sum_j K_{\ell,j}(\gamma_\ell)$ and $\widetilde{W}_\ell = \sum_j K_{\ell,j}(\hat{\gamma}_\ell) = \widetilde{W}_\ell - (\hat{\gamma}_\ell - \gamma_\ell) \sum_j H_j$.

Note that by (6), $\mathbb{E}(H_j) = 0$. By Proposition 1 and (6), we have $\mathbb{E}(\widetilde{W}_\ell) = \frac{n}{L}$. Furthermore, because $\widetilde{W}_\ell$ is a sum of i.i.d. random variables, we have that as $n \to \infty$,

$$\check{\mathbf{A}}_n = \sqrt{\frac{L}{n}} \left( \widetilde{W}_1 - \frac{n}{L}, \widetilde{W}_2 - \frac{n}{L}, \ldots, \widetilde{W}_L - \frac{n}{L} \right) \xrightarrow{d} \mathcal{N}_L(0, \check{\Omega}), \tag{12}$$

where for each $\ell, \ell^* \in \{1, \ldots, L\}$,

$$\check{\Omega}_{\ell,\ell*} = L\mathrm{Cov}\left\{ K_{\ell,j}(\gamma_\ell), K_{\ell*,j}(\gamma_{\ell*}) \right\}.$$

Therefore,

$$\check{U}_{n,L} = \check{\mathbf{A}}_n^\mathsf{T} \check{\mathbf{A}}_n \xrightarrow{d} \chi_{\check{\Omega}}^2.$$

Next, we will show that the actual statistic $\widetilde{U}_{n,L}$ is close to $\check{U}_{n,L}$, and that the estimated variance matrix is also close to $\check{\Omega}$. We start with noting that the estimator $\hat{\gamma}_\ell$ from linear regression is close to the optimal choice $\gamma_\ell$ defined in (7): by the Central Limit Theorem, $\hat{\gamma}_\ell = \gamma_\ell + \mathcal{O}_p(1/\sqrt{n})$. And thus

$$\widetilde{W}_\ell = \sum_{j=1}^{n} w_j \left( \mathbb{1}\{\ell_j = \ell\} - \hat{\gamma}_\ell a(X_j, Z_j, V_j) \right) + n\hat{\gamma}_\ell \mathbb{E}_{\mathcal{T}}[a(X, Z, V)]$$

$$= \sum_{j=1}^{n} w_j \mathbb{1}\{\ell_j = \ell\} - \hat{\gamma}_\ell \left( \sum_{j=1}^{n} w_j a(X_j, Z_j, V_j) - n\mathbb{E}_{\mathcal{T}}[a(X, Z, V)] \right)$$

$$= \sum_{j=1}^{n} w_j \mathbb{1}\{\ell_j = \ell\} - \gamma_\ell \left( \sum_{j=1}^{n} w_j a(X_j, Z_j, V_j) - n\mathbb{E}_{\mathcal{T}}[a(X, Z, V)] \right) + \mathcal{O}_p(1)$$

$$= \widetilde{W}_\ell + \mathcal{O}_p(1).$$

The second-to-last line is because $\hat\gamma_\ell = \gamma_\ell + \mathcal{O}_p(1/\sqrt{n})$ and the terms inside the parenthesis, $\sum_j H_j$, is a sum of $n$ independent mean-zero random variables.

Therefore, together with (12), by Slusky's Theorem, we have that

$$\widetilde{\mathbf{A}}_n = \sqrt{\frac{L}{n}}\left(\widetilde{W}_1 - \frac{n}{L}, \widetilde{W}_2 - \frac{n}{L}, \ldots, \widetilde{W}_L - \frac{n}{L}\right) \xrightarrow{d} \mathcal{N}_L(0, \breve\Omega),$$

and thus,

$$\widetilde{U}_{n,L} = \widetilde{\mathbf{A}}_n^\mathsf{T}\widetilde{\mathbf{A}}_n \xrightarrow{d} \chi^2_{\breve\Omega}.$$

We will work on sample covariance matrix now. Recall that the sample covariance matrix $\widetilde\Omega_n = \frac{L}{n}(\mathbf{W} - \frac{1}{L} \cdot \mathbf{1}_{L\times n})(\mathbf{W} - \frac{1}{L} \cdot \mathbf{1}_{L\times n})^\mathsf{T}$, where $\mathbf{W}_{\ell,j} = w_j \cdot \left[\mathbb{1}\{\ell_j = \ell\} - \hat\gamma_\ell a(X_j, Z_j, V_j)\right] + \hat\gamma_\ell \mathbb{E}_\mathcal{T}\left[a(X, Z, V)\right] = K_{\ell,j}(\hat\gamma_\ell)$. Let's start with $\mathbf{W}\mathbf{W}^\mathsf{T}$. For any $\ell, \ell* \in \{1, \ldots, L\}$,

$$\begin{aligned}
(\mathbf{W}\mathbf{W}^\mathsf{T})_{\ell,\ell*} &= \sum_j K_{\ell,j}(\hat\gamma_\ell)K_{\ell*,j}(\hat\gamma_{\ell*}) \\
&= \sum_j \left(K_{\ell,j}(\gamma_\ell) - (\hat\gamma_\ell - \gamma_\ell)H_j\right)\left(K_{\ell*,j}(\gamma_\ell) - (\hat\gamma_{\ell*} - \gamma_{\ell*})H_j\right) \\
&= \sum_j K_{\ell,j}(\gamma_\ell)K_{\ell*,j}(\gamma_{\ell*}) - (\hat\gamma_\ell - \gamma_\ell)\sum_j H_j K_{\ell*,j}(\gamma_{\ell*}) - (\hat\gamma_{\ell*} - \gamma_{\ell*})\sum_j H_j K_{\ell,j}(\gamma_\ell) \\
&\qquad\qquad + (\hat\gamma_\ell - \gamma_\ell)(\hat\gamma_{\ell*} - \gamma_{\ell*})\sum_j H_j^2 \\
&= \sum_j K_{\ell,j}(\gamma_\ell)K_{\ell*,j}(\gamma_{\ell*}) + \mathcal{O}_p(\sqrt{n})
\end{aligned}$$

Therefore, by the law of large numbers,

$$\frac{L}{n}(\mathbf{W}\mathbf{W}^\mathsf{T})_{\ell,\ell*} = \frac{L}{n}\sum_j K_{\ell,j}(\gamma_\ell)K_{\ell*,j}(\gamma_{\ell*}) + \mathcal{O}_p(1/\sqrt{n}) = L\mathbb{E}\left[K_{\ell,1}(\gamma_\ell)K_{\ell*,1}(\gamma_{\ell*})\right] + \mathcal{O}_p(1/\sqrt{n}).$$

Similarly, for $\mathbf{W}\mathbf{1}^\mathsf{T}$, we have that for any $\ell, \ell* \in \{1, \ldots, L\}$,

$$(\mathbf{W}\mathbf{1}^\mathsf{T})_{\ell,\ell*} = \sum_j K_{\ell,j}(\hat\gamma_\ell) = \sum_j K_{\ell,j}(\gamma_\ell) - (\hat\gamma_\ell - \gamma_\ell)H_j = \sum_j K_{\ell,j}(\gamma_\ell) + \mathcal{O}_p(\sqrt{n}).$$

Therefore, again by the law of large numbers,

$$\frac{L}{N}(\mathbf{W}\mathbf{1}^\mathsf{T})_{\ell,\ell*} = \frac{L}{N}\sum_j K_{\ell,j}(\gamma_\ell) + \mathcal{O}_p(1/\sqrt{n}) = L\mathbb{E}\left[K_{\ell,j}(\gamma_\ell)\right] + \mathcal{O}_p(1/\sqrt{n}) = 1 + \mathcal{O}_p(1/\sqrt{n}).$$

Combining the above results gives,

$$\begin{aligned}
\widetilde\Omega_{n,\ell,\ell*} &= \frac{L}{n}\left[(\mathbf{W} - \frac{1}{L} \cdot \mathbf{1}_{L\times n})(\mathbf{W} - \frac{1}{L} \cdot \mathbf{1}_{L\times n})^\mathsf{T}\right]_{\ell,\ell*} \\
&= L\mathbb{E}\left[K_{\ell,1}(\gamma_\ell)K_{\ell*,1}(\gamma_{\ell*})\right] - L\mathbb{E}\left[K_{\ell,j}(\gamma_\ell)\right]\mathbb{E}\left[K_{\ell*,j}(\gamma_{\ell*})\right] + \mathcal{O}_p(1/\sqrt{n}) \\
&= L\,\mathrm{Cov}\left[K_{\ell,1}(\gamma_\ell), K_{\ell*,1}(\gamma_{\ell*})\right] + \mathcal{O}_p(1/\sqrt{n}) \\
&= \breve\Omega_{\ell,\ell*} + \mathcal{O}_p(1/\sqrt{n}).
\end{aligned}$$

Therefore, $\widetilde\Omega_n \xrightarrow{p} \breve\Omega$.

To summarize, we have that

$$\widetilde{U}_{n,L} = \widetilde{\mathbf{A}}_n^\mathsf{T}\widetilde{\mathbf{A}}_n \xrightarrow{d} \chi^2_{\breve\Omega}, \quad \text{and} \quad \theta_{\widetilde\Omega_n,\alpha} \xrightarrow{p} \theta_{\breve\Omega,\alpha}.$$

Therefore,

$$\mathbb{P}(\text{Algorithm 2 rejects}) = \mathbb{P}(\widetilde{U}_{n,L} \geqslant \theta_{\widetilde\Omega_n,\alpha}) \to \mathbb{P}(\chi^2_{\breve\Omega} \geqslant \theta_{\breve\Omega,\alpha}) = \alpha.$$

### A.3 Proof of Theorem 2

Similar to the proof of Theorem 1, we define

$$\widetilde{W}_\ell = \sum_{j=1}^n w_j \left( \mathbb{1}\{\ell_j = \ell\} - \gamma_\ell a(X_j, Z_j, V_j) \right) + n\gamma_\ell \mathbb{E}_\mathcal{T}[a(X, Z, V)],$$

$$K_{\ell,j}(\gamma) = w_j \left( \mathbb{1}\{\ell_j = \ell\} - \gamma a(X_j, Z_j, V_j) \right) + \gamma \mathbb{E}_\mathcal{T}[a(X, Z, V)], \text{ and}$$

$$H_j = w_j a(X_j, Z_j, V_j) - \mathbb{E}_\mathcal{T}\left[ a(X, Z, V) \right].$$

Therefore, we have $\widetilde{W}_\ell = \sum_j K_{\ell,j}(\gamma_\ell)$ and $\widetilde{W}_\ell = \sum_j K_{\ell,j}(\hat{\gamma}_\ell) = \widetilde{W}_\ell - (\hat{\gamma}_\ell - \gamma_\ell) \sum_j H_j$.

We know from the literature that $\gamma_\ell$ is the optimal choice of $\gamma$ and thus $\mathrm{Var}\left[ \widetilde{W}_\ell \right] \leqslant \mathrm{Var}\left[ W_\ell \right]$. We will then move on to show that $\mathrm{Var}\left[ \widetilde{W}_\ell \right]$ is close to $\mathrm{Var}\left[ \widetilde{W}_\ell \right]$ and thus asymptotically no greater than $\mathrm{Var}\left[ W_\ell \right]$.

To this end, note that

$$\begin{aligned}
\mathrm{Var}\left[ \widetilde{W}_\ell \right] &= \mathrm{Var}\left[ \widetilde{W}_\ell - (\hat{\gamma}_\ell - \gamma_\ell) \sum_j H_j \right] \\
&= \mathrm{Var}\left[ \widetilde{W}_\ell \right] + 2 \,\mathrm{Cov}\left[ \widetilde{W}_\ell, (\hat{\gamma}_\ell - \gamma_\ell) \sum_j H_j \right] + \mathrm{Var}\left[ (\hat{\gamma}_\ell - \gamma_\ell) \sum_j H_j \right] \\
&\leqslant \mathrm{Var}\left[ \widetilde{W}_\ell \right] + 2 \sqrt{\mathrm{Var}\left[ \widetilde{W}_\ell \right]} \sqrt{\mathbb{E}\left[ \left( (\hat{\gamma}_\ell - \gamma_\ell) \sum_j H_j \right)^2 \right]} + \mathbb{E}\left[ \left( (\hat{\gamma}_\ell - \gamma_\ell) \sum_j H_j \right)^2 \right].
\end{aligned}$$

But we also know from the proof of Theorem 1 that $\hat{\gamma}_\ell - \gamma_\ell \overset{p}{\to} 0$. Then, because of the bounded fourth moment assumption, by the Dominated Convergence Theorem, we have that

$$\frac{1}{n} \mathbb{E}\left[ \left( (\hat{\gamma}_\ell - \gamma_\ell) \sum_j H_j \right)^2 \right] \to 0.$$

Therefore,

$$\limsup_{n \to \infty} \frac{1}{n} \left( \mathrm{Var}\left[ \widetilde{W}_\ell \right] - \mathrm{Var}\left[ \widetilde{W}_\ell \right] \right) \leqslant 0.$$

Finally, we note that $\mathrm{Var}\left[ W_l \right] = \Omega(n)$, and hence

$$\limsup_{n \to \infty} (\mathrm{Var}\left[ \widetilde{W}_l \right] / \mathrm{Var}\left[ W_l \right]) \leqslant 1.$$

## B  Additional Simulation Results

### B.1  Running time

All experiments run on a Macbook Pro 2022 M2.

**Artificial dataset**: Regarding running time for one iteration including density ratio estimation and $X|Z$ model fitting (on average), csPCR took 5.12s, csPCR(pe) took 14.95s, IS method took 1.5s, PCR took 1.25s.

**Real-world application**: Regarding running time for one test procedure, csPCR took 3.41s, csPCR(pe) took 11.32s, IS method took 0.81s.

### B.2  Finding optimal hyperparameter $L$

We find the optimal $L$ value for the testing algorithm by performing numerical simulations, evaluating its Type-I error control and power. We adopt the same numerical simulation setup as in the main text Section 4. We first choose $a_\mathcal{S} = 1$ and $a_\mathcal{T} = 0$ and also fix $\beta = 1$ to compare the Type-I error rate for different choice of $L$ of the csPCR method. We perform experiments with both true density ratio and estimated density ratio. The results are shown in Table 2.

Table 2: Type-I Error Rates at Different Levels of L of csPCR Method

| L | 2 | 3 | 5 | 10 | 15 | 20 |
|---|---|---|---|---|---|---|
| True Density Ratio | 0.05125 | 0.05000 | 0.04575 | 0.03675 | 0.02825 | 0.02425 |
| Estimated Density Ratio | 0.04620 | 0.05025 | 0.04425 | 0.03905 | 0.02725 | 0.02175 |

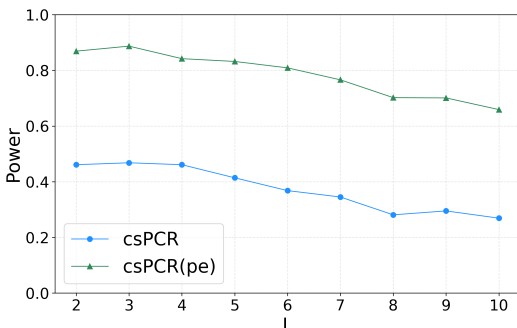

Figure 5: Comparison of statistical power of the three methods as the the parameter $L$ varies.

We also test the power of the csPCR and csPCR(pe) method with different choices of $L$ value. We choose $a_{\mathcal{S}} = 0$ and $a_{\mathcal{T}} = 2$ and fix $\beta = 2$.

As Table 2 and Figure 5 shows, as $L$ value increases, the csPCR method become more conservative with more tight Type-I error control and lower power. We can observe that when we set $L = 3$, the csPCR method can achieve most stable Type-I error rate control and also highest power empirically. Therefore, in our simulation experiments and real world data experiments, we fix $L = 3$.

### B.3 Role of effective sample size

We notice a series of work in measuring the effective sample size (ESS) of importance weight or sampling in the statistical computation literature, e.g., [Martino, et al, 2017] and others. Among them, one of the most common ways is to use the ratio $n_{eff} = \frac{\left(\sum_{i=1} w_i\right)^2}{\sum_{i=1} w_i^2}$ to approximate the ESS. When the covariate shift between the source and target becomes stronger, the variance of the importance weight $w_i$ tends to be large and $n_{eff}$ will become smaller, which can result in lower power. Our power enhancement method based on control variate could potentially alleviate this issue with properly specified control functions.

In the simulation study, we varied only $\mu_z$, the mean of the confounding variables $Z_{\mathcal{T}}$. A higher $\mu_z$ signifies a stronger covariate shift between the source and target populations. From Figure 6, it is evident that as $\mu_z$ increases, the Effective Sample Size (ESS) required significantly decreases, while the power of the csPCR method concurrently declines. These results suggest that increasing covariate shift leads to a reduction in ESS and a corresponding decrease in statistical power.

### B.4 Instability of the Importance Resampling (IS) method

In this section, we will use numerical simulations to ilustrate that the performance if the IS method is subject to the resample size heavily. IS method performs resampling without replacement and typically has to sample a much smaller subset (theoretically, in the order of $o(\sqrt{n})$) of the source data to approximate the target. Consequently, the power of IS is substantially lower than our approach. If the resample size of IS is overly increased, it may fail to control the Type-I error due to excessive similarity between the resampled data and the original source data.

To further illustrate, we conducted additional experiments with varied resample sizes in IS to assess its effect on Type-I error control and power. From Figure7. one can observe that IS starts to show high Type-I error inflation when its resample size increases to 400 but still shows much lower power

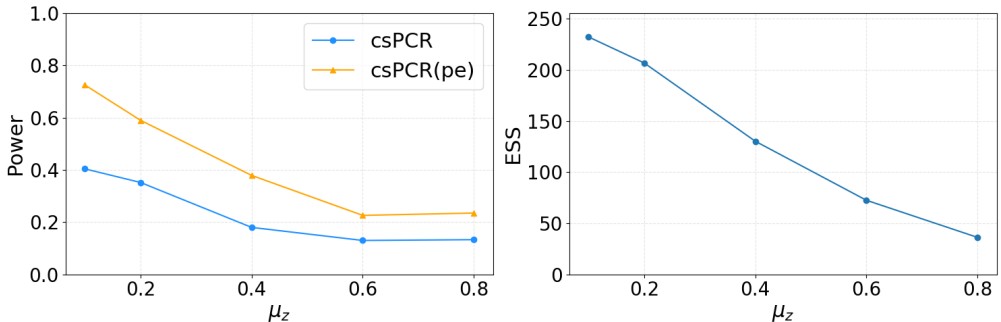

Figure 6: The left panel shows the comparison of statistical power of csPCR and csPCR(pe) method as the covariate shift gets stronger. The right panel illustrates how the Effective Sample Size(ESS) changes as covariate shift scale becomes larger.

(by around 0.4) than our method with this resample size (or even larger ones). This indicates that our method achieves better statistical efficiency than IS (DRPL).

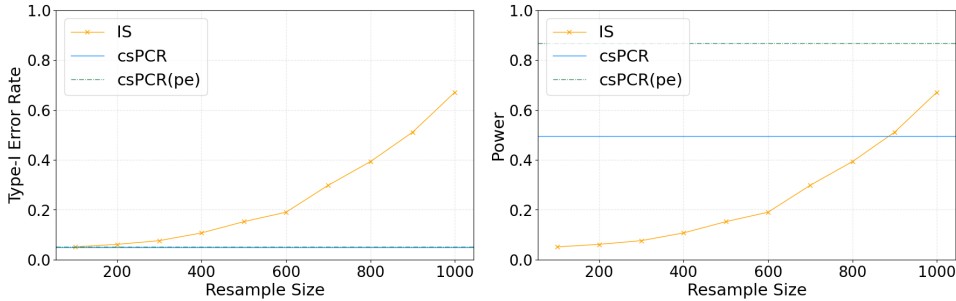

Figure 7: Detailed comparison of Type-I error rate and power of csPCR and the IS method. With the source sample size $n_s = 1000$, we gradually increase the resample size for the IS method from 100 to 1000. The two horizontal lines represent the Type-I error rate and power, respectively, of the csPCR and csPCR(pe) methods (they do not change with the tuning of IS).

### B.5   Choice of test statistic

In this section we explore the effect of test statistics on the algorithm performance. The main principle of choosing the test statistic is to characterize the conditional dependency between $X$ and $Y$ under the alternative hypothesis. The test statistic $YX$ may not be the optimal choice and that using $(Y - \hat{E}[Y \mid Z])(X - E[X \mid Z])$ could remove the confounding effect of $Z$.

Inspired by this, we used $Y(X - E[X|Z])$ as the test statistic to conduct additional simulations. As illustrated in Figure 8, we find that $Y(X - E[X|Z])$ and $YX$ produce nearly the same power for both csPCR and csPCR(pe) with the change of effect size.

## C   Real-World Application

The specific medication indicated by the treatment variable $X$ includes Ritonavir, Bamlanivimab, Casirivimab-Imdevimab, Remdesivir, Ritonavir Nirmatrelvir, Sotrovimab, Bamlanivimab Etesevimab. For simplicity, $X = 1$ indicates any of these specific medication and $X = 0$ otherwise.

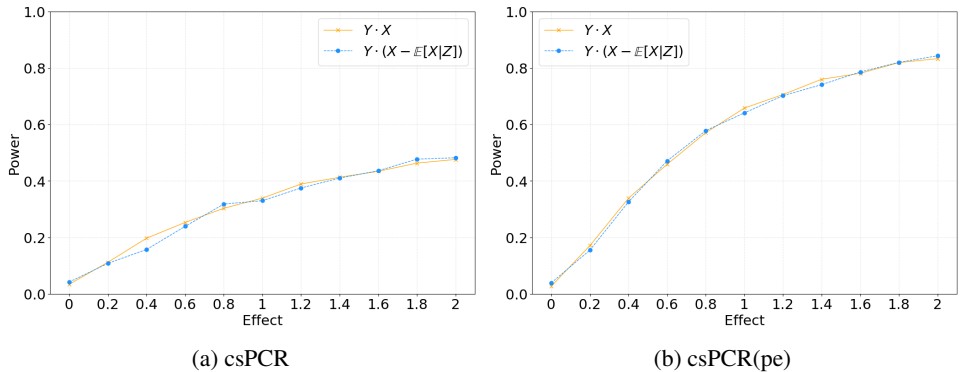

(a) csPCR

(b) csPCR(pe)

Figure 8: Power against effect size for csPCR and csPCR(pe) with two different test statistics $XY$ and $(X - \mathrm{E}[X \mid Z])Y$. We observe that the power is very similar with the two different test statistics.

## C.1 Different outcome

In our real data experiment part, the outcome variable $Y$ is defined as mortality within 90 days since hospital admission due to COVID-19. In addition, we also analyzed mortality within 30 days since hospital admission. As shown in Table 3, both csPCR and csPCR(pe) methods give significant results, aligning with biomedical literature.

Table 3: $p$-values of different methods on COVID-19 dataset (mortality 30

| Method | csPCR | csPCR(pe) |
|---|---|---|
| $p$-value | 0.029 | 0.013 |

