# OpenReview forum: "Covariate Shift Corrected Conditional Randomization Test"
_NeurIPS.cc/2024/Conference — NeurIPS 2024 poster_

### Official Review · Reviewer_REEU · 2024-07-04

**Soundness:** 4
**Presentation:** 4
**Contribution:** 3
**Rating:** 7
**Confidence:** 4

**Summary:**

The paper introduces a novel approach to addressing covariate shift in conditional independence tests. By leveraging importance weights and the control variates method,the paper proposes Covariate Shift Corrected Pearson Chi-squared Conditional Randomization (csPCR) test, which maintains asymptotic Type-I error control. In addition, a power enhancement is proposed, which reduces the variance in the test statistic and thus improve the power towards alternatives. The methodology is validated both theoretically and empirically, demonstrating superior performance in simulation studies and practical application to a COVID-19 treatment dataset, highlighting its efficacy in real-world scenarios.

**Strengths:**

Originality
The paper presents original contributions by introducing a new approach to conditional independence testing under covariate shift. This method combines the idea of conditional randomization test, importance weighting, the control variates technique.

Quality
The work is of high quality, demonstrated through rigorous statistical theories on the validity and power, and comprehensive empirical validations.

Clarity
The paper is clearly written and well-organized. Each section has a clear message and readers can quickly grasp the main idea.

Significance
I consider this work as a significant contribution to the field of conditional independence testing and covariate shift. The method is easy-to-implement, and is more powerful than existing methods as shown in numerical experiments.

**Weaknesses:**

The paper does not leave serious technical weaknesses, but I am concerned about the limited applicability of this method in real-world problems. Based on my understanding, one would test for conditional independence directly on the target distribution if the outcomes in the target population were easy to collect. So csPCR is useful only when the outcomes are costly to collect. This questions several application examples mentioned in the paper. First, in the college admission example (from Line 37-53), why don't the economists just collect the college admission results from the target population and then conduct the vanilla PCR? Typically, the target outcomes have to be collected for followup analysis after the conditional independence test. Second, Section 5 introduces a real-world application of this method on Covid-19 pandemic data, where the source and the target data set are segmented based on time. Given such a sequential data collecting process, it is unclear whether the iid assumption still holds in the source and target distribution. See Questions for more details.

Despite of the concerns above, I do believe that this method is useful for certain genetic and biomedical problems, where the outcomes are difficult to collect, but other covariates (such as SNPs) are relatively easy. Therefore, I suggest the authors clarify the application examples a bit by discussing more reasonable applications with some references.

**Questions:**

1. Page 5, Algorithm 1: Besides the variance issues considered in Algorithm 2, it is clear that the choice of test statistic in Eq (4) also affects the power. Can you comment on how to choose the test statistic? In the numerical experiment, the test statistic is specified to be the product of $Y$ and $X$, which makes sense as it measures the "correlation" between Y and X. However, $Z$ is not included in the test statistic, which seems to decrease its power. Have you tried any other test statistics? Do difference choices affect the power significantly? For instance, one alternative is to use $(Y - \beta_y Z)(X - \beta_x Z)$, where $\beta_y and \beta_x$ are regression coefficients for $Y\sim Z$ and $X\sim Z$ on an independent data set from source distribution.

2. Page 4, Line 170-177: The paper argues that they used PCR as the baseline test because PCR is more powerful than the vanilla CRT. But on the other hand, CRT is also better than PCR in the sense that it has finite-sample type I error control, rather than asymptotic type I error control. It's worth mentioning this point as well. While I notice that the simulations in the paper shows satisfactory type I error control, the type I errors may get inflated if, for example, the data have heavy tails.

3. Page 9, Section 9: Relating to the weaknesses, have you run any tests to check the iid assumption on the Covid-19 data set?

**Limitations:**

The papers discuss about the limitations of their method under model misspecifications in Section 4, pointing out that the powerful enhancement can disappear when there is a full nonlinear component in the model. The authors have addressed the limitations adequately.

---

> ### Author Rebuttal · Authors · 2024-08-07
>
> We extend our sincere thanks for your reviewing work and insightful comments. Please see our responses to your comments and questions as below.
>
> $\textbf{Weaknesses 1}$: In the college admission example, why don't the economists just collect the college admission results from the target population?
>
> Response: In the college admission example, large-scale data on SAT scores is more easily accessible through high schools, exam preparation schools, or the College Board (the organization that administers the SAT exam). Conversely, college admission results are more difficult and costly to obtain, as they require individual-level surveys. Furthermore, students might be reluctant to disclose whether they were rejected in such surveys.
>
> Following your suggestion, we also plan to add another example about clinical study. Suppose we are interested in testing the treatment effect of some drug $X$ on some long-term outcome $Y$ such as five-year survival. In this case, researchers collect $V$ as some early endpoint surrogate that can be measured within a short term, usually a few weeks or months post treatment, e.g., tumor response rate in cancer treatment; see [e.g., VanderWeele 2013].
>
> VanderWeele, Tyler J. "Surrogate measures and consistent surrogates." Biometrics 69.3 (2013): 561-565.
>
>
> $\textbf{Weaknesses 2}$: Given such a sequential data collecting process of the COVID data, it is unclear whether the iid assumption still holds.
> & $\textbf{Question 3}$: Have you run any tests to check the iid assumption?
>
> Response: The period spanning from January 2020 to November 30, 2021, in our source dataset captures the early waves of the COVID-19 pandemic, including the original strain and early variants such as Alpha and Beta. After November 30, 2021, our target dataset includes admissions during subsequent waves dominated by variants like Delta and Omicron. The segmentation of the dataset at November 30, 2021, is intentional to account for significant shifts in virus characteristics, public health policies, and medical treatments, while within source or target, the distribution of these covariates is assumed to be consistent, reflecting the stabilization of public health responses and medical treatments specific to early/later variants and the increased coverage of vaccinations. In addition, we plan to run rigorous tests (e.g., Kolmogorov-Smirnov test) to examine this i.i.d assumption within the source and target upon the acceptance of the paper.
>
>
> $\textbf{Question 1}$: Besides the variance issues considered in Algorithm 2, it is clear that the choice of test statistic in Eq (4) also affects the power. Can you comment on how to choose the test statistic?
>
> Response: The main principle of choosing the test statistic is to characterize the conditional dependency between $X$ and $Y$ under the alternative hypothesis. We agree that $YX$ may not be the optimal choice for the test statistic and that using $(Y-\hat{E}[Y\mid Z])(X - E[X\mid Z])$ could remove the confounding effect of $Z$. Inspired by this, we used $Y(X - E[X|Z])$ as the test statistic to conduct additional simulations. The results are presented in supplementary Figure R3. We find that $Y(X - E[X|Z])$ and $YX$ produce nearly the same power for both csPCR and csPCR(pe).
> We did not use $(Y-\hat{E}[Y\mid Z])(X - E[X\mid Z])$ because estimating $\hat{E}[Y\mid Z]$ requires sample splitting to estimate $\hat{E}[Y\mid Z]$ with some hold-out sample (otherwise, the theoretical Type-I error control of the PCR test cannot be guaranteed).
> An alternative strategy is to get an estimate of $P(Y\mid X,Z)$ on some hold-out training data as mentioned above, then naturally use $\log{P(Y\mid X,Z)}-\log{P(Y\mid Z)}$ as the test statistic [Tansey et al., 2022]. This could increase the ability to capture nonlinear dependence, but the sample splitting will generally cause a loss of power. The surrogate or auxiliary $V$ in our case could potentially help this hold-out training procedure and alleviate the power loss issue. We plan to add simulations on this upon the acceptance of our paper.
> Tansey, Wesley, et al. "The holdout randomization test for feature selection in black box models." Journal of Computational and Graphical Statistics 31.1 (2022): 151-162.
>
>
> $\textbf{Question 2}$: Comparison of PCR with vanilla CRT?
>
> Response:  You are perfectly correct that PCR does not achieve exact Type-I error control. Meanwhile, we would like to point out that unlike other asymptotic inference approaches, PCR and csPCR are less prone to issues with heavy-tailed data, as they rely on the binary indicator variable of each subject to construct the chi-squared statistics. Nevertheless, small sample sizes could still cause Type-I error inflation in csPCR. We note that the PCR paper [Javanmard and Mehrabi, 2021] introduced a finite-sample Type-I error control version of PCR, which is achieved using concentration inequalities. Their strategy could be naturally incorporated with the current csPCR to achieve better (exact) Type-I control. We will add a discussion on this extension.
> Additionally, we agree that our statement regarding PCR being more powerful than CRT was overly broad. A more precise statement would be that PCR can handle some more challenging alternatives not addressed well by the vanilla CRT, as demonstrated in [Javanmard and Mehrabi, 2021]. In our numerical studies, we used the PCR construction on all benchmarks for a fairer comparison, and we found that our method outperforms the IS method with the vanilla CRT as well.
>
> $\textbf{References}$
>
> Javanmard, Adel, and Mohammad Mehrabi. "Pearson chi-squared conditional randomization test." arXiv preprint arXiv:2111.00027 (2021).

---

> > ### Comment · Reviewer_REEU · 2024-08-08
> >
> > Thanks for the clarification. I don't have further concerns.

---

### Official Review · Reviewer_r237 · 2024-07-11

**Soundness:** 3
**Presentation:** 3
**Contribution:** 3
**Rating:** 7
**Confidence:** 3

**Summary:**

This paper proposes a new variation of CRT to be applied in the presence of covariate shifts. The paper presents a method and one extension for each, with higher power. Then, the authors present some needed theoretical results and finish with experiments.

**Strengths:**

- The paper proposes a theoretically correct approach for testing CI under covariate shift;
- The paper presents their algorithms and theoretical results;
- The paper had convincing experiments;

**Weaknesses:**

- It would be interesting if the authors could estimate the full density ratio in their simulations (possibly in a high-d scenario) and then compare their results with the IS approach;
- The paper does not conduct any experiment where Type-I error control is shown in a real dataset, e.g., like for example in [1,2].

References

[1] Pogodin, R., Schrab, A., Li, Y., Sutherland, D. J., & Gretton, A. (2024). Practical Kernel Tests of Conditional Independence. arXiv preprint arXiv:2402.13196.

[2] Maia Polo, Felipe, Yuekai Sun, and Moulinath Banerjee. "Conditional independence testing under misspecified inductive biases." Advances in Neural Information Processing Systems 36 (2023): 58577-58612.

**Questions:**

- How can we understand the role of the effective sample sizes (weight imbalance) in the effectiveness of your method?
- Could you elaborate more on the surrogate variables? I am familiar with the literature on CI testing, but I haven't seen the presence of these variables in other works.

**Limitations:**

The authors comment on limitations.

---

> ### Author Rebuttal · Authors · 2024-08-07
>
> We extend our sincere thanks for your reviewing work and insightful comments. Please see our responses to your comments and questions as below.
>
> $\textbf{Weakness 1}$: It would be interesting if the authors could estimate the full density ratio in their simulations (possibly in a high-d scenario) and then compare their results with the IS approach.
>
> Response: We have added additional experiments with the full density ratio estimated, as shown in Figure R2 of the supplementary PDF file (attached to the global rebuttal response). We chose a high-dimensional setting where the dimension of $Z$ is 50. We fit high-dimensional regression to estimate the joint density ratio $e(X, Z, V)$ and the conditional model $X \sim Z$ with the unlabeled data.
> The results show that when the sample size is larger than 800, all three methods have good Type-I error control. With a relatively small sample size for estimation, the IS method and the power enhancement version of our method retain good Type-I error rate control, but the vanilla csPCR method may have an inflated Type-I error rate (e.g., when $n_e = 400$, the Type-I error rate of csPCR is 0.108, while the IS method and csPCR(pe) retain 0.052). The statistical power follows a similar pattern as presented in the paper (csPCR(pe) > csPCR > IS) but is uniformly lower for all three methods because of the lower estimation accuracy (e.g., when $\beta = 2$, the power of csPCR(pe) is 0.833, while in our original experiments it is 0.867).
>
> $\textbf{Weakness 2}$: The paper does not conduct any experiment where Type-I error control is shown in a real dataset, e.g., like for example in [1,2].
>
> Response: We propose the following steps to validate Type-I error control in the real dataset. (1) We will adjust the cutoff date for segmentation to an earlier point, expanding the target dataset. For example, moving the cutoff date from November 30, 2021, to a mid-2021 date, increases the sample size in the target dataset while still capturing significant changes in the pandemic landscape. With the expanded dataset, we will rerun our csPCR test and calculate the empirical Type-I error rate. This involves comparing the test results against the true outcomes in the expanded target dataset. (2) Additionally, we will generate multiple permuted datasets by randomly shuffling Y while keeping X and Z unchanged. For each permuted dataset, we will apply the csPCR test and record the results to calculate the empirical Type-I error rate.
>
>
> $\textbf{Question 1}$: How can we understand the role of the effective sample sizes (weight imbalance) in the effectiveness of your method?
>
> Response: We notice a series of work in measuring the effective sample size (ESS) of importance weight or sampling in the statistical computation literature, e.g., [Martino, et al, 2017] and others. Among them, one of the most common ways is to use the ratio $n_{eff}=(\sum_{i=1}w_i)^2 / \sum_{i=1}w_i^2$ to approximate the ESS. When the covariate shift between the source and target becomes stronger, the variance of the importance weight $w_i$ tends to be large and $n_{eff}$ will become smaller, which can result in lower power. Our power enhancement method based on control variate could potentially alleviate this issue with properly specified control functions. We plan to carry out additional simulation studies on the relationship between the power of csPCR and the effective sample size (affected by the degree of covariate shift) in the camera-ready version of the paper upon acceptance.
>
> $\textbf{Question 2}$: Could you elaborate more on the surrogate variables? I am familiar with the literature on CI testing, but I haven't seen the presence of these variables in other works.
>
> Response: A surrogate or silver standard label is a variable that is more feasible and accessible than $Y$ in data collection and can be viewed as a noisy measure of $Y$. In clinical trials, $Y$ is often a longer-term outcome; in such settings, surrogates are measures that can predict the effect of a treatment on the longer-term outcome $Y$. These surrogates can be biomarkers or clinical parameters measured relatively quickly, usually within a few weeks or months of starting treatment. For example, in clinical trials, tumor response rate is often used as a surrogate for overall survival, and blood pressure is commonly used as a surrogate for cardiovascular events such as heart attacks. Surrogate variables are also commonly used in environmental studies and economics.
>
> Regarding conditional independence testing, Li and Liu (2023) study how surrogate variables can improve the robustness of the Conditional Randomization Test (CRT). They propose a method called Maxway CRT, which leverages knowledge of the distribution of $Y|Z$ to enhance the robustness of the CRT. Surrogate variables are extremely helpful in learning the distribution of $Y|Z$ because there is usually much more data on surrogates than on the outcome variable.
>
> Finally, we want to emphasize that without the surrogate variable, even if there is a shift in the distribution of $X$, the original CRT on the source data remains valid for the target population, provided that the distribution of $Y | X, Z$ is assumed to be the same in both the source and target populations.
>
> $\textbf{References}$
>
> Martino, Luca, Víctor Elvira, and Francisco Louzada. "Effective sample size for importance sampling based on discrepancy measures." Signal Processing 131 (2017): 386-401.
>
> Li, Shuangning, and Molei Liu. "Maxway CRT: improving the robustness of the model-X inference." Journal of the Royal Statistical Society Series B: Statistical Methodology 85.5 (2023): 1441-1470.

---

> ### Comment · Reviewer_r237 · 2024-08-09
>
> Thank you for your reply!
>
> - W1: Thank you for your new experiment! (this suggestion is related to the ESS question because in high-d we expect the ESS to be lower)
> - W2: I think the idea is good. When shuffling Y, you will need to do that within each value of Z, correct? If your Z is continuous, you would probably need to use some binning.
> - Q1: My question was related to the paper you mentioned here by Martino et al. In that paper, they argue that the ESS can be defined as a variance ratio and the implications of that are clear in the IS literature. I was wondering if you could extract some more meaningful relationships here as well.
> - Q2: Thank you for the detailed explanation! I would probably try to input some introduction on the surrogate variable in the abstract since you mention it but it might not be clear you are going to use them throughout the paper (you are working in this specific setup, which is not the same as the original CRT setup).
>
> I have increased my score.

---

> > ### Author Response · Authors · 2024-08-10
> >
> > We express sincere thanks to your insightful comments and positive recognition on our work.
> >
> > $\textbf{W1}$: Thanks for pointing out that an increasing dimension tends to cause higher variational importance weights and lower ESS. We will follow this to design our additional simulation on power v.s. ESS (changing with the variance of importance weights).
> >
> > $\textbf{W2}$: Thanks! When shuffling $Y$, we think both (i) marginal permutation and (ii) conditioning on (the full set or a subset of) Z will cause Y and X to be independent conditional on Z. Thus, we will try both setups and you are correct that we can use binning if Z is continuous and relatively high-dimensional.
> >
> > $\textbf{Q1}$: Thanks for the question! We think the effective sample size of our csPCR estimator (without control variate) can be defined in a similar way as their paper, i.e., the original source sample size * the ratio between the variances of (i) and (ii), where (i) stands for the unweighted indicator on target (which one would obtain if they could observe and use the same amount of labeled sample on target for PCR), and (ii) stands for the weighted indicator variable on source.
> >
> > This ratio can be shown to be smaller or equal to 1, with the “=” holding only when there is no covariate shift between the source and target. Thus, covariate shift will cause loss of the effective sample size to our method and larger degree of covariate shift tends to induce larger variance of the weighted estimator on source and larger loss of the ESS. This interpretation is quite analog to that in IS literature. Further, we can write down the ESS for our power enhanced (PE) estimator and apply our Theorem 2 to show that it is larger than the ESS of csPCR without PE.
> >
> > $\textbf{Q2}$: Thanks for the suggestion and we will highlight more on the specific setup with surrogate in the abstract and introduction section!

---

### Official Review · Reviewer_MwvS · 2024-07-12

**Soundness:** 3
**Presentation:** 3
**Contribution:** 2
**Rating:** 5
**Confidence:** 3

**Summary:**

This paper introduces the Covariate Shift Corrected Pearson Chi-squared Conditional Randomization (csPCR) test, designed to address covariate shift in conditional independence testing. The csPCR method incorporates importance weights and employs the control variates method to enhance test power and reduce variance in the statistical analysis. Theoretical contributions demonstrate that csPCR controls Type-I error asymptotically. Empirical validations through simulation studies and a real-world application assessing COVID-19 treatment effectiveness showcase the method's effectiveness.

**Strengths:**

1. **Theoretical contribution:** The paper makes a theoretical contribution by addressing the covariate shift in conditional independence testing.

2. **Methodological Innovation:** Introduction of the Covariate Shift Corrected Pearson Chi-squared Conditional Randomization (csPCR) test that incorporates importance weights and control variates method to manage covariate shifts effectively.

3. **Empirical Validation:** Extensive simulation studies and real-world application (assessment of COVID-19 treatment on 90-day mortality) demonstrate the practical efficacy and superior power of the proposed csPCR test over traditional methods.

**Weaknesses:**

1. **Dependence on Accurate Estimations of Density Ratio:** The csPCR test's performance is critically dependent on the accurate estimation of density ratios. This dependence could pose significant challenges in practical scenarios characterized by limited, noisy, or high-dimensional data, where reliable density ratio estimation becomes inherently difficult.

2. **Clarity on the Advantage Over Resampling Methods:** The paper does not explicitly clarify the advantages of the csPCR test over simpler resampling-based methods that also utilize estimated density ratios. While the csPCR integrates density ratios directly into the test statistic calculation and employs variance reduction techniques, it remains crucial for the authors to demonstrate why these features offer substantial improvements over traditional methods that adjust the sample weights during resampling. The discussion should address whether the complexity of csPCR provides tangible benefits, such as improved error rates or robustness in more varied practical applications, compared to potentially simpler resampling approaches.

**Questions:**

- Could the authors provide a detailed comparison between the csPCR test and traditional resampling methods that also use estimated density ratios? This would further strengthen the paper
- The paper mentions the use of control variates to reduce variance introduced by importance weights. Could the authors discuss how this approach compares to variance reduction techniques used in resampling methods?
- Could the authors elaborate on any theoretical limitations or potential failure modes of the csPCR method?
- Are there additional empirical validations planned or underway to test the csPCR method across more varied datasets?

**Limitations:**

The authors are encouraged to add a separate limitation section in either the main paper or appendix.

---

> ### Author Rebuttal · Authors · 2024-08-07
>
> $\textbf{Weakness}$ 1: Dependence on Accurate Estimations of Density Ratio \& $\textbf{Question 3}$: Theoretical Limitations?
>
> Response: We believe our method's limitation lies mainly when the model-X assumption fails, i.e., when the distribution of covariates cannot be accurately learned. In such settings, Type-I error inflation may occur. The robustness of CRT and PCR tests under this condition has been studied extensively [Berrett et al., 2020; Javanmard and Mehrabi, 2021; Li and Liu, 2023]. Theoretical upper bounds for Type-I error inflation have been given; moreover, simulation studies show that CRT and PCR tests are usually more robust than these bounds suggest.
>
> For our csPCR test, we theoretically believe we can show similar results to those in Section 6 of [Javanmard and Mehrabi, 2021]. Empirically, we conducted simulation studies (some included in our submission, plus new ones) when the covariate distribution is estimated from data:
> Our studies include two scenarios: (i) one has the knowledge of $P(X\mid Z)$ and needs to estimate $P(V \mid X, Z)$; (ii) one has to estimate the full $P(X, V, Z)$ without any knowledge. For (i), please refer to Figure 3 for our numerical results showing that even with moderate sample sizes used for learning $P(V \mid X, Z)$ (e.g., $n_e=500$, equal to the labeled sample size used for testing), the csPCR test still maintains good Type-I error control (nearly no inflation above 0.05). For the more challenging scenario (ii), we have added additional experiments with the full distribution density (or density ratio) estimated using the unlabeled samples. The results can be found in Figure R2 of the supplementary PDF file (attached to the global rebuttal response). In this case, our csPCR(pe) approach can still achieve Type-I error control at $n_e=400$ as well as good power, and the vanilla csPCR shows proper Type-I error control at $n_e=700$, which is still not large compared to the testing sample size of 500.
>
> $\textbf{Weakness 2}$: Advantage Over Resampling Methods
> & $\textbf{Question 1}$: Comparison with traditional resampling methods?
>
> Response: First, we would like to clarify that we are benchmarking against the specific DRPL resampling method (referred to as IS in our paper) proposed in [Nikolaj et al., 2023]. This approach is proposed for the general purpose of testing under covariate shift and is, to our best knowledge, the only existing strategy for conditional independence testing under covariate shift. It is essentially different from traditional bootstrap resampling or importance sampling procedures. Specifically, IS performs resampling without replacement and typically has to sample a much smaller subset (theoretically, in the order of $o(\sqrt{n})$) of the source data to approximate the target. Consequently, the power of IS is substantially lower than our approach. If the resample size of IS is overly increased, it may fail to control the Type-I error due to excessive similarity between the resampled data and the original source data.
> To further illustrate, we conducted additional experiments with varied resample sizes in IS to assess its effect on Type-I error control and power; see Figure R1 in the supplementary PDF file (attached to the global rebuttal response). One can observe that IS starts to show high Type-I error inflation when its resample size increases to 400 but still shows much lower power (by around 0.4) than our method with this resample size (or even larger ones). This indicates that our method achieves better statistical efficiency than IS (DRPL). We also find that similar results hold regardless of whether the density ratio and model of $X$ are known or estimated.
>
> $\textbf{Question 2}$: Variance reduction techniques used in resampling methods?
>
> Response: As highlighted in our response to your Question 1, the IS method is essentially different from traditional resampling methods. The authors in [Nikolaj et al., 2023] have not proposed any variance reduction methods such as control variate, and we do not see a natural way to accommodate control variate in their framework. Therefore, we do not see a feasible way to make a direct comparison.
>
> $\textbf{Question 4}$: Additional empirical validations?
>
> Response: We plan to add the following new experiments with the real-world data:
> 1. Expansion of the target dataset. We will adjust the cutoff date for segmentation to an earlier point, expanding the target dataset. For example, moving the cutoff date from November 30, 2021, to a mid-2021 date, increases the sample size in the target dataset while still capturing significant changes in the pandemic landscape.
> 2. With the COVID data, we will generate multiple permuted datasets by randomly shuffling the outcome variable Y while keeping the treatment variable X and covariates Z unchanged. This process ensures that the relationship between Y and X, given Z, is broken, simulating the null hypothesis for use to test the robustness of our method in Type-I error control.
> 3. Different outcomes. In addition to readmission, we will evaluate different outcomes such as mortality. Specifically, we will analyze mortality within 30 and 90 days of hospital admission due to COVID-19.
>
>
> $\textbf{References}$
>
> Berrett, Thomas B., et al. "The conditional permutation test for independence while controlling for confounders." Journal of the Royal Statistical Society Series B: Statistical Methodology 82.1 (2020): 175-197.
>
> Javanmard, Adel, and Mohammad Mehrabi. "Pearson chi-squared conditional randomization test." arXiv preprint arXiv:2111.00027 (2021).
>
> Li, Shuangning, and Molei Liu. "Maxway CRT: improving the robustness of the model-X inference." Journal of the Royal Statistical Society Series B: Statistical Methodology 85.5 (2023): 1441-1470.
>
> Thams, Nikolaj, et al. "Statistical testing under distributional shifts." Journal of the Royal Statistical Society Series B: Statistical Methodology 85.3 (2023): 597-663.

---

> ### Author Response · Authors · 2024-08-07
>
> We extend our sincere thanks for your reviewing work and insightful comments. Please see our responses to your comments and questions as below in the rebuttal.

---

> > ### Comment · Reviewer_MwvS · 2024-08-14
> >
> > Thanks for your detailed response.  Most major concerns have been solved. I would like to keep my original rating.

---

> ### Comment · Area_Chair_oa13 · 2024-08-11
> **Reminder to engage in the discussion**
>
> Hello reviewer MwvS. The reviewer-author discussion period is between Aug 7-13 (AoE). Please read the authors’ rebuttal, see if it addresses your questions, and engage in a discussion with the authors. You are strongly encouraged to read the official reviews posted by other reviewers. Some of your questions may have already been answered. **You must acknowledge the authors’ rebuttal and give a reason for why it did or did not address your concerns. If the rebuttal does not change your rating, please also state the reason.** Thank you for your services.

---

### Official Review · Reviewer_G2Ah · 2024-07-15

**Soundness:** 2
**Presentation:** 2
**Contribution:** 2
**Rating:** 4
**Confidence:** 1

**Summary:**

This paper addresses the issue of conditional independence under covariate shift. The authors' goal is to test the conditional independence for causal inference in the target data. The authors can use the source data whose distribution is potentially different from that of the target data. For this problem, the authors propose using the covariate-shift adaptation method and develop a method for testing the null hypothesis.

**Strengths:**

This paper addresses the issue of conditional independence under covariate shift, which I find to be an intriguing problem. However, I was unable to comprehend the underlying assumptions of this problem, making it difficult to proceed with the reading.

I understand the objective of this manuscript to be as follows:
1. The authors aim to test the null hypothesis $H_0: X \perp Y \mid Z$ for the target data (Here, I denote the independence by $\perp$).
2. There is a possibility that $X \perp Y \mid Z$ does not hold simultaneously for both the source and target data.
3. To test the null hypothesis for the target data, the authors employ an algorithm adapted to covariate shift.
My primary concern is whether the assumption "there is a possibility that $X \perp Y \mid Z$ does not hold simultaneously for both the source and target data" is justified in the first place.

The authors assume that, in order to use the covariate shift adaptation algorithm,
- $Y \mid X, Z, V$ is the same in both the source and target data.
Under this assumption, is it not the case that the situation where $X \perp Y \mid Z$ holds in one dataset and does not hold in the other cannot exist?

$X \perp Y \mid Z$ implies that $p(y, x \mid z) = p(y \mid x, z) p(x \mid z) = p(y \mid z) p(x \mid z)$, meaning that $p(y \mid x, z) = p(y \mid z)$.
Given that we assume $Y \mid X, Z, V$ to be the same in both the source and target data, does this not imply that whether $X \perp Y \mid Z$ holds cannot differ between the source and target data?

Due to this ambiguity, I was unable to further evaluate the paper. I would like the authors to clarify this point so that I can proceed with the evaluation.

**Weaknesses:**

See above.

**Questions:**

See above.

**Limitations:**

See above.

---

> ### Author Rebuttal · Authors · 2024-08-07
>
> We extend our sincere thanks for your reviewing work and important comments. We believe your main confusion lies in the existence of V and the fact that $P(V \mid X, Z)$ can be different between the source and target. This can possibly cause the situation that $H_0: X \perp Y \mid Z$ does not hold simultaneously on the source and target (note that V is an auxiliary feature not included in the hypothesis of our primary interest). For illustration, consider the following simplified example:
>
> Let $X \sim N(0, 1)$ and $Z = X + \epsilon_z$ on both source and target populations, where $\epsilon_z$ is an independent noise term. For the source, let $V_{\mathcal{S}} = -X_{\mathcal{S}} + \epsilon_v$, while on the target, let $V_{\mathcal{T}} = \epsilon_v$, where $\epsilon_v$ is a noise term independent of X and Z. On both source and target, let $Y = X + Z + V + \epsilon$, corresponding to our assumption that $P(Y \mid X, Z, V)$ holds the same between the source and target.
>
> In this case, one can derive that (i) on the source, $Y_{\mathcal{S}} = Z + \epsilon_v + \epsilon$; on target, $Y_{\mathcal{T}} = X + Z + \epsilon_v + \epsilon$. Thus, $X \perp Y \mid Z$ holds on the source but not on the target, which underscores the importance of our setup and method.
>
> A more general data generation setup can be found in Figure 2 of our paper. In this diagram, $V$ could be interpreted as a mediator or early endpoint surrogate seen in real-world studies. Taking clinical trials as an example, suppose we are interested in testing the treatment effect of some drug $X$ on some long-term outcome $Y$ such as five-year survival, adjust for some baseline confounder $Z$. In this case, researchers collect $V$ as some early endpoint surrogate that can be measured within a short term, usually a few weeks or months post treatment, e.g., tumor response rate in cancer treatment. Thus, $V$ is easier to collect and also informative to $Y$. Then it is reasonable to assume that $P(Y\mid X,Z,V)$ is shared by the source and target populations while $P(V\mid X,Z)$ has a distributional shift between the two populations [Kallus and Mao, 2020]. Please find other real-world examples in our response to Weakness 1 from Reviewer REEU. We hope this clarification addresses your concerns and enables you to proceed with the evaluation. Thank you!
>
> $\textbf{References}$
>
> Kallus, Nathan, and Xiaojie Mao. "On the role of surrogates in the efficient estimation of treatment effects with limited outcome data." arXiv preprint arXiv:2003.12408 (2020).

---

> ### Comment · Reviewer_G2Ah · 2024-08-11
> **Re: Rebuttal by Authors**
>
> I appreciate the authors' reply, which has deepened my understanding of your contributions.
>
> It seems I might have caused some misunderstanding with my previous question. Here is what I intended to ask:
>
> Objective of this study: To investigate whether conditional independence holds in the target data.
>
> Assumption in this study: Covariate shift = meaning that P(Y | X, Z, V) remains unchanged between the source data and the target data.
>
> Under this situation, wouldn't testing only the source data suffice to achieve the objective, given the assumption?
>
> I wanted to confirm whether I misunderstood the problem setting.

---

> > ### Author Response · Authors · 2024-08-12
> >
> > Thank you for your reply and clarification! You are correct that our goal is to test $X \perp Y \mid Z$ on the target data, with the assumption that $P(Y \mid X, Z, V)$ remains consistent between the source and target domains. However, even if $P(Y \mid X, Z, V)$ is the same for both the source and target, $P(V \mid X, Z)$ can still differ between them. As we demonstrated in our example in the rebuttal, this difference in $P(V \mid X, Z)$ led to $X \perp Y \mid Z$ holding true on the source data but not on the target. If we only test on the source data, this discrepancy can significantly inflate the Type-I error rate, as illustrated in Figure 1 of our paper.
> > Please Let us know if you have any further concerns.

---

> ### Comment · Reviewer_G2Ah · 2024-08-13
> **Re: Official Comment by Authors**
>
> Thank you for your response. I have gained a deeper understanding of the authors' contributions.
>
> However, to be honest, I am not entirely clear about the motivation behind this study. If the authors assume that the conditional distributions are equal between the source and target data, wouldn't it suffice to test only for the source data? I am not convinced about the necessity of concerning with the Type 1 error in test for the target data. In the other words, it is unclear for me why we conduct test for the target data.
>
> I also have doubts about the claim that the Type 1 error changes. I think this is due to covariate shift, but if we consider covariates as non-stochastic, wouldn't the Type 1 error remain unchanged?
>
> My background is actually in economics. While the authors provide examples from economics, in social science data analysis, it is common to estimate models using only the source data and then conduct counterfactual simulations on the target data under the assumption that the conditional distributions are equal. That is, it is enough to care only about the model on the source data, and we can transfer the results under the assumption that the conditional distribution is invariant. Given these typical analysis procedures, I find the problem setting somewhat difficult to understand.
>
> I do not oppose acceptance, but I have some reservations about the problem setting, so I will keep my current score. In the future, since this problem setting may not be intuitively accepted, it might be worthwhile to focus more on justifying the problem setting itself more rather than on the technical issue of controlling the Type 1 error.

---

### Author Rebuttal · Authors · 2024-08-07

We sincerely thank all reviewers and chairs for their feedback. We have addressed each of the reviewers' points individually and have included a PDF with additional experiments. The first of the experiments demonstrated that our proposed csPCR method has more stable Type I error rate control and higher power without the need for tuning the resample size compared to the IS method. The second experiment shows that when estimating the full density ratio, the vanilla csPCR may suffer from slight Type I error inflation, while other results show similar patterns to our original experiments. The third experiment shows that with the different choice of test statistic $Y(X - E[X|Z])$, both the Type I error rate and power do not change significantly.

---

### Decision · Program_Chairs · 2024-09-25

**Decision:**

Accept (poster)

**Comment:**

This paper addresses the problem of conditional independence testing under covariate shifts. The goal is to determine whether $p(y|x,z) = p(y|z)$ (i.e., whether X and Y are conditionally independent given Z) on the target data, by making use of another source dataset. The main challenge arises because there can be covariate shifts between the source and the target. The paper introduces importance weights into the Pearson Chi-Squared Conditional Randomization (PCR) test and makes use of the control variates method to address the potential covariate shifts. It is theoretically proven that the new test is consistent (asymptotically has maximum test power), and asymptotically has a well-controlled false positive rate.

**Exclude an official review from Reviewer G2Ah**

Reviewer G2Ah misunderstood the problem setting and could not go through the submission to provide a full review. In particular, in the paper, it is assumed that $P(Y|X, Z, V)$ remains the same from source to target data. The misunderstanding was that this would mean $P(Y|X,Z)$ is also the same and so the test can be conducted on the source data to draw conclusions on the target data. This statement is NOT true due to the presence of $V$. This misunderstanding was resolved during the reviewer discussion period with the AC. This official review is negative because of a misunderstanding. The AC decided to exclude this review.

The authors are strongly advised to emphasize this point more. In particular, it may be useful to include the derivation to arrive at $P(Y|X,Z)=\int P(Y|X,Z,V)P(V|X,Z)\thinspace dV$ in the paper.


**Overall review**

The remaining three reviewers noted the originality of the method in this specific setting (REEU), and strong theoretical contribution (r237) with convincing experiments (MwvS). A common point raised is that the method depends on the accuracy of the estimated density ratio (for importance weighting) (MwvS, r237) which may limit applicability of the method in real problems (REEU). The authors sufficiently addressed this concern by providing more empirical results in the rebuttal.

Recommendation: accept.

The authors are strongly suggested to add more results to questions raised by the reviewers e.g.,generate multiple permuted datasets on the Covid data experiment, evaluate different outcomes such as mortality, etc.